# An innate granuloma eradicates an environmental pathogen using *Gsdmd* and *Nos2*

Carissa K. Harvest[1,2,3], Taylor J. Abele [1,2], Chen Yu [4], Cole J. Beatty [1,4], Megan E. Amason [1,2,3], Zachary P. Billman[1,2,3], Morgan A. DePrizio[1,2], Fernando W. Souza[1,2,5], Carolyn A. Lacey [1,2], Vivien I. Maltez[3], Heather N. Larson [1,2], Benjamin D. McGlaughon[3], Daniel R. Saban[1,4], Stephanie A. Montgomery[6] & Edward A. Miao [1,2,5,7] ✉

Granulomas often form around pathogens that cause chronic infections. Here, we discover an innate granuloma model in mice with an environmental bacterium called *Chromobacterium violaceum*. Granuloma formation not only successfully walls off, but also clears, the infection. The infected lesion can arise from a single bacterium that replicates despite the presence of a neutrophil swarm. Bacterial replication ceases when macrophages organize around the infection and form a granuloma. This granuloma response is accomplished independently of adaptive immunity that is typically required to organize granulomas. The *C. violaceum*-induced granuloma requires at least two separate defense pathways, gasdermin D and iNOS, to maintain the integrity of the granuloma architecture. This innate granuloma successfully eradicates *C. violaceum* infection. Therefore, this *C. violaceum*-induced granuloma model demonstrates that innate immune cells successfully organize a granuloma and thereby resolve infection by an environmental pathogen.

*Chromobacterium violaceum* is a Gram-negative bacterium found in freshwater sediment and soils. This colorful bacterium produces a violet pigment and encodes a type III secretion system (T3SS) that is similar to the invasion-associated T3SS found in *Salmonella* species[1]. *C. violaceum* uses this T3SS to invade and replicate in nonphagocytic cells[2,3]. Immunocompetent people are likely exposed to *C. violaceum*, but almost never develop symptomatic infection. Conversely, immunocompromised patients can be highly susceptible to *C. violaceum* infection. For example, patients with chronic granulomatous disease, who have defects in the phagocyte NADPH oxidase, are highly susceptible to disseminated *C. violaceum* infection with a mortality rate of around 55%[1,4]. Concomitantly, mice lacking the NADPH oxidase

succumb to low-dose *C. violaceum* infection within one day. In addition, we previously discovered that resistance to *C. violaceum* requires the T3SS-sensing NLRC4 inflammasome, caspase-1, and gasdermin D[3,5]. Mice deficient in these pyroptosis-inducing genes succumb to very low-dose infection[3]. Therefore, *C. violaceum* has potent virulence potential conferred by its T3SS that is fully counteracted by a functional innate immune system.

This defense requires NADPH oxidase and inflammasomes, however, herein we demonstrate a much more complex innate immune defense that must be orchestrated to eradicate *C. violaceum*. Large, distinct lesions form in the liver of wild-type C57BL/6 mice during *C. violaceum* infection[5,6]. Nevertheless, these wild-type mice never

[1]Department of Integrative Immunobiology, Duke University School of Medicine, Durham, NC, USA. [2]Department of Molecular Genetics and Microbiology, Duke University School of Medicine, Durham, NC, USA. [3]Department of Microbiology and Immunology, University of North Carolina at Chapel Hill, Chapel Hill, NC, USA. [4]Department of Ophthalmology, Duke University School of Medicine, Durham, NC, USA. [5]Department of Cell Biology, Duke University School of Medicine, Durham, NC, USA. [6]Department of Pathology, University of North Carolina at Chapel Hill, Chapel Hill, NC, USA. [7]Department of Pathology, Duke University School of Medicine, Durham, NC, USA. ✉e-mail: edward.miao@duke.edu

display visible clinical signs and survive the infection. In this manuscript, we discover that these liver lesions are organized granulomas that form to eradicate this environmental bacterial pathogen.

Granulomas are an organized aggregation of immune cells that surround a persistent stimulus, the origin of which can be either infectious or noninfectious. Granuloma responses are initiated during infection by pathogens across diverse classes of microorganisms, including bacteria, parasites, fungi, and viruses[7–17]. A defining characteristic of a granuloma is the recruitment and organization of inflammatory macrophages into a layer that surrounds the infection[7–11,14,15,18]. These inflammatory macrophages can differentiate into different subtypes depending on the pathogen. Sometimes these differentiation states are visibly identifiable in histologic sections (e.g., epithelioid, giant multinuclear, and foamy macrophages), but in other granulomas, specific morphologic changes are less apparent. Numerous other immune cell types can be present, most prominently T cells, that activate macrophages and are normally thought to be essential to organize the granuloma architecture[7,8,10,14,18–22]. Fibroblasts are also typically present and deposit layers of collagen in the granuloma. Granulomas associated with different infections can have diverse histologic architecture[7–17]. For example, some granulomas contain necrotic cores at their centers while others do not, and some even contain high numbers of neutrophils[8]. Regardless of the specific histologic architecture of each granuloma, all contain, and are defined by, organized zones of macrophages.

Granulomas often form around persistent infectious agents, and granulomas often fail to clear such infections[7,10,19,22,23]. These pathogens can survive within the granuloma, leading many to conclude that the granuloma response is a last resort used against an infection that cannot be cleared. Thus, granulomas are often considered a way to restrain an infectious agent and prevent its dissemination, but without the ability to eradicate the microbe. Actually, individual granulomas, either between different infected animals or even multiple granulomas in the same animal, can be heterogeneous—some granulomas clear the infection whereas other granulomas have progressive infection containing viable organisms that are not able to be cleared[24]. Given the diverse clinical outcomes and the lack of understanding as to what drives the formation of heterogeneous granulomatous responses, more models are needed to study the range of the granuloma responses, particularly mouse models that leverage advanced immunological technologies[25,26].

Herein, we examine the granuloma response to *C. violaceum* from the initiating events through the clearance of the bacterium and provide definitive evidence for the beneficial role of pyroptotic proteins during the granuloma response.

## Results

### *C. violaceum* triggers formation of a necrotic granuloma
Intraperitoneal infection of wild-type C57BL/6 J mice with *C. violaceum* ($1 \times 10^2$ colony-forming units; CFUs) does not result in lesions[3]. However, when a higher infectious dose ($1 \times 10^4$ CFUs) is used, WT mice form macroscopic lesions, specifically in the liver (Fig. 1a)[5,6]. Lesions were evenly distributed and typically 1–2 mm in diameter at 5 days post infection (dpi) (Fig. 1b). Working with a board-certified veterinary pathologist (S.A.M.), we investigated the histologic morphology by hematoxylin and eosin (H&E) staining and consistently observed three distinct layers in every lesion (Fig. 1c, colored boxes). First, the center was characterized by dark hematoxylin (purple) staining of amorphous material that lacked defined cell borders and features, consistent with necrotic debris (Fig. 1c, green box, and 1d). Surrounding this necrotic core was the second layer, composed of degenerative hepatocytes that retained cell borders but displayed faded hematoxylin chromatin staining in the nuclei, or lacked a visible nucleus altogether. In this layer, the cytoplasm stained brightly eosinophilic (pink), and hepatocyte boarders remained apparent (Fig. 1c, e; blue arrows), indicative of

coagulative necrosis[27]. These coagulative necrotic hepatocytes were also sporadically present within the necrotic core (Fig. 1d; green arrows).

The third layer, exterior to the layer of coagulative necrosis, was predominantly composed of cells with a large, oval nucleus that contained open chromatin which is consistent with activated macrophages (Fig. 1c, f; orange arrowhead). Within this "macrophage zone", scattered fibroblasts were present, identified by their fusiform shape and a nucleus that was dark and elongated (Fig. 1f; carrot). Immediately exterior to the macrophage zone are viable hepatocytes at 5 dpi (Fig. 1f; asterisk). The distinct, organized zone of macrophages surrounding each lesion is a defining feature of a granuloma, with the presence of a necrotic core subcategorizing them as necrotic granulomas. Altogether, this model displays striking granulomas at the 5 dpi timepoint.

### A granuloma forms in response to one bacterium
To determine whether each granuloma was initiated by a single bacterium, we infected mice with a 1:1 ratio of WT (violacein positive) and *ΔvioA* mutant (violacein negative) *C. violaceum* strains, dissected single granulomas, and determined the number of CFUs and color of the colonies (Fig. 1g and S1a). The majority of single granulomas contained only one *C. violaceum* strain (Fig. 1g and S1a). This result suggests that granulomas are seeded by a single bacterium. In addition, WT and *ΔvioA* strains seeded equivalent numbers of granulomas with similar burdens per granuloma (Fig. 1g), indicating that the violacein pigment is dispensable for liver infection[28]. This suggests that a single bacterium replicates to a median of $5.4 \times 10^6$ CFUs in each granuloma by 3 dpi. Clearly, the innate immune response fails to halt *C. violaceum* replication within the first 3 dpi.

### *C. violaceum* first infects hepatocytes
We next investigated the early hours of infection to understand granuloma development. *C. violaceum* infects hepatocytes using its T3SS[2], a cell type in which we were unable to detect caspase-1 protein[3], suggesting that this niche is not defended by caspase-1. We visualized *C. violaceum* by immunohistochemical staining at 12 hours post infection (hpi) and observed bacteria within hepatocytes (Fig. 1h). We observed multiple bacteria within each hepatocyte, suggesting that the bacteria were replicating by 12 hpi. Bacteria were also observed in immune cells (Fig. S1b). Bacterial burdens in the liver increased by ~100-fold between 6 hpi and 24 hpi (Fig. 1i). This 1.5 hour doubling time in vivo is remarkably similar to the 1 hour doubling time in vitro in broth (Fig. S1c). To visualize how this hepatocyte niche would develop in the absence of neutrophils, we depleted neutrophils with anti-GR1 and then infected the mice. This revealed large microcolonies of *C. violaceum* at 24 hpi, some of which were the size of a single hepatocyte, while others had the appearance of multiple hepatocytes infected in a cluster (Fig. 1h and S1d).

### Neutrophil swarming fails to eradicate *C. violaceum*
At 24 hpi, a transition occurs when larger microabscesses appear, comprised of a swarm of neutrophils identified morphologically by their multilobulated nucleus, and confirmed by Ly6G staining (Fig. 1j). Macrophages were absent from these microabscesses by morphological identification, also verified by CD68 staining (Fig. 1j and S1e). *C. violaceum* staining showed bacteria only within the neutrophil swarm and absent from surrounding hepatocytes (Fig. 1j). Bacteria were tightly clustered within individual neutrophils (Fig. 1j). In contrast, the spleen had low burdens at 6 hpi, and burdens were cleared by 18 hpi (Fig. 1i) and remained sterile at 3 dpi[3]. In the liver, bacterial replication slows after 24 hpi but does not stop, with a 14.5 hour doubling time (Fig. S1c). Thus, neutrophils fail to clear the bacteria and *C. violaceum* continues to replicate within microabscesses in the liver.

Although the neutrophils failed to clear the bacteria, reactive oxygen species (ROS) are required to slow bacterial replication over

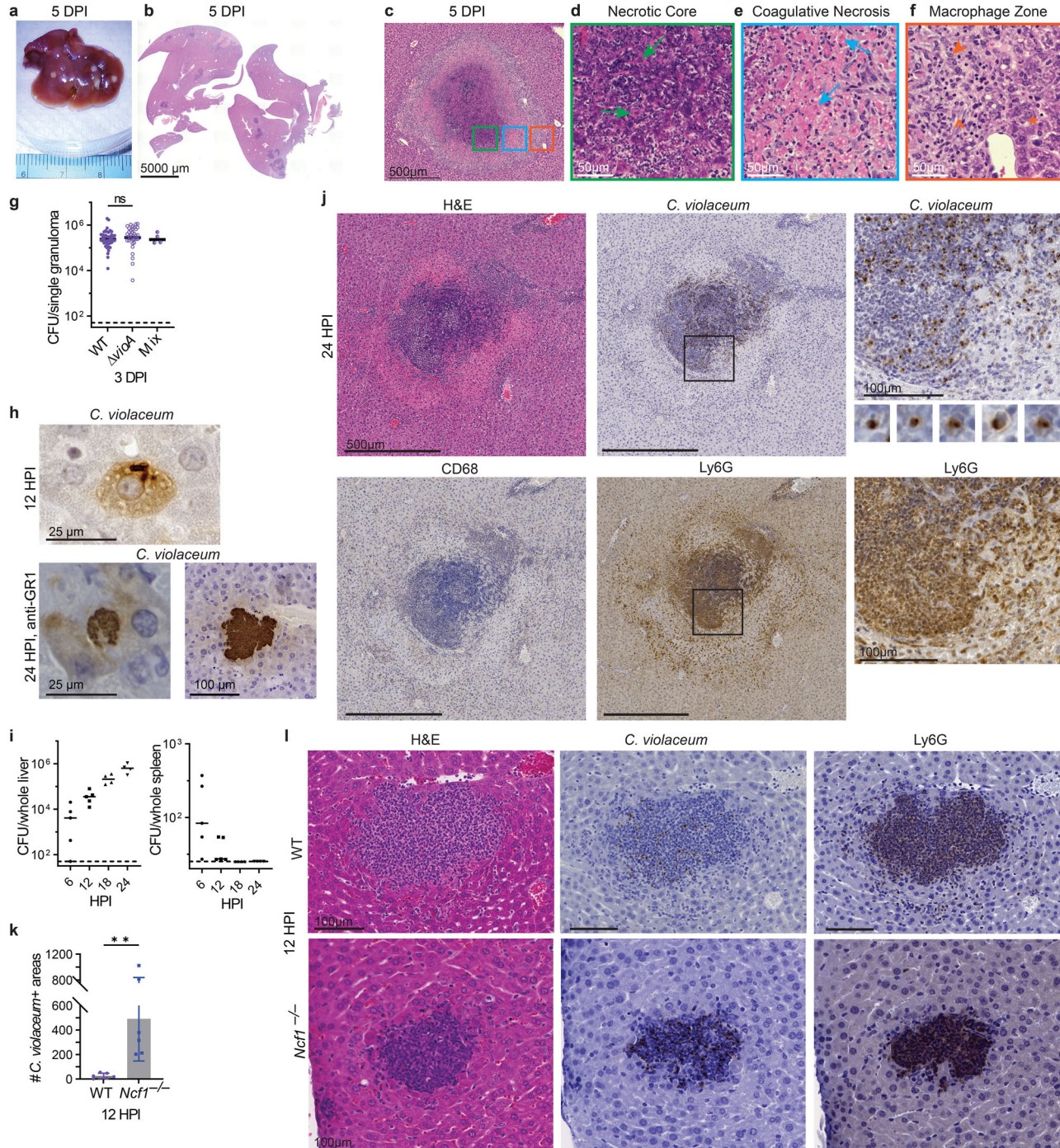

**Fig. 1 | *C. violaceum* replicates despite a neutrophil swarm. a–l** Mice were infected with 10⁴ CFU WT *C. violaceum*, strains indicated. Dashed line, limit of detection; solid line, median. **a–f** Gross pathology and H&E staining of WT livers 5 dpi. Representative of 10 experiments, each with 3–4 mice, each with multiple granulomas per section. Arrows, coagulative necrotic hepatocyte; arrowhead, macrophage; carrot, fibroblast; asterisk, healthy hepatocytes. **g** Inoculum was a 1:1 mixture of WT and *ΔvioA C. violaceum*. 83 individual granulomas were dissected 3 dpi, bacterial burdens determined, and scored for color. Data combined from two experiments. N.s., not significant by Kruskal–Wallis test. **h** Visualization of *C. violaceum* within a hepatocyte at 12 hpi by IHC and within hepatocytes in anti-GR1-treated mice at 24 hpi by IHC. For anti-GR1 depletion infections, mice were treated with either anti-GR1 or isotype control at day −1 and 0

and harvested at 24 hpi, with 4 mice per treatment group. Data representative of two experiments. **i** Bacterial burdens in the liver and spleen at indicated timepoints. Representative of 2 experiments; each point is a single mouse. **j** Serial sections of WT liver stained by H&E or indicated IHC markers 1 dpi. Representative of 5 experiments, each with 4 mice, each with multiple granulomas per section. **k** Quantitation of *C. violaceum*-positive stained areas per section 12 hpi in WT and *Ncf1⁻/⁻* mice. Data combined from two experiments, each with three mice per genotype. Bar represents mean ± SD. **p = 0.0078, by unpaired two-tailed *t* test. **l** Serial sections of WT or *Ncf1⁻/⁻* livers stained by H&E or indicated IHC markers 12 dpi. Representative of 2 experiments, each with 3–4 mice per genotype.

the first 24 hpi. Myeloid cells generate ROS in the phagosome using the phagocyte NADPH oxidase, which includes p47$^{phox}$ encoded by *Ncf1*. *Ncf1*$^{-/-}$ mice succumb to even 100 CFUs of *C. violaceum* within 24 hpi[3]. *Ncf1*$^{-/-}$ mice had an increased number of *C. violaceum*-positive areas in the liver at 12 hpi (Fig. 1k). These areas ranged from a few neutrophils to sizable neutrophil swarms, albeit smaller than those seen in WT mice (Fig. 1l). Again, *C. violaceum* staining clustered tightly within the neutrophil swarm in *Ncf1*$^{-/-}$ mice (Fig. 1l). Thus, although NADPH oxidase functions to limit the seeding of lesions by *C. violaceum*, neutrophils nevertheless fail to eradicate *C. violaceum* which continues to grow within the neutrophil swarm of WT mice.

### A granuloma forms around the infected neutrophil swarm

Because neutrophils fail to eradicate *C. violaceum*, a more effective immune response is needed. By 3 dpi, the microabscesses have grown (Fig. 2a). The neutrophils observed at 1 dpi now form the bulk of the necrotic core at 3 dpi, identified by Ly6G staining (Fig. 2b, c). We observed a zone of coagulative necrotic hepatocytes surrounding the necrotic core (Fig. 2b). *C. violaceum* staining was observed in the necrotic core, but not inside the sparse coagulative necrotic hepatocytes in the core (Fig. 2d; carrots). At 3 dpi, we observed a thin macrophage zone identified by morphology and confirmed with CD68 staining (Fig. 2b, c). Bacterial staining was essentially absent in the macrophage zone, except rare areas where neutrophils were infiltrating the periphery of the granuloma (Fig. 2b, c). Further, no *C. violaceum* staining was observed in the healthy hepatocytes outside the microabscess (Fig. 2b, c). At this timepoint, the architecture and presence of macrophages indicates the transition from a microabscess to the formation of an early granuloma, which can be subcategorized into a necrotic suppurative granuloma (pyogranuloma)[8].

At 5 dpi, the granuloma response has matured with a distinct and thick macrophage zone confirmed with pronounced CD68 staining (Fig. 2b, c), while neutrophil staining at the periphery became sparser (Fig. 2b, c). Concomitantly, we identified fibroblasts by morphology and colocalized collagen deposition, both within the macrophage zone (Fig. 2b, c). *C. violaceum* staining was still predominantly within the necrotic core (overlapping with Ly6G); notably, all the bacterial staining was contained within the granuloma (Fig. 2b, c). At 7 dpi the granuloma macrophage zone had become even thicker while the necrotic core appears smaller than at 3 and 5 dpi (Fig. 2a–c). At this timepoint, the coagulative necrosis zone persists and collagen staining is more prominent (Fig. 2b). In the granulomas of other animal models, epithelioid macrophages are present and express E-cadherin to form tight junctions, functioning to help wall off the pathogen[29]. However, we did not observe E-cadherin staining colocalizing with macrophages in the *C. violaceum*-induced granuloma (Fig. S1f). In our model, granuloma macrophages also lacked foamy and multinucleated morphology. This *C. violaceum*-induced granuloma model demonstrates the ability of inflammatory macrophages to organize a granuloma that successfully prevents bacterial dissemination.

At 14 dpi, most granulomas were visibly smaller, although rare granulomas remain large in occasional mice (Fig. 3a; arrow). In a typical 14 dpi granuloma the macrophage zone remained prominent, while the necrotic core became even smaller; in some granulomas the necrotic core was completely absent (Fig. 3b and S1g; arrows). The coagulative necrotic zone remained prominent, although the cellular borders of the coagulative necrotic hepatocytes were no longer distinct. The macrophage zone remained prominent and began to infiltrate the coagulative necrotic zone (Fig. 3b). At earlier timepoints, *C. violaceum* staining and CD68 staining do not overlap, with *C. violaceum* staining predominantly in the center of the granuloma (Fig. 2b, c). However, surprisingly at 14 dpi these stains now overlap in the macrophage zone (Fig. 3b and S1h). This staining pattern suggests that dead bacterial antigens have been phagocytosed by macrophages in the process of removing the necrotic core. This is consistent with most

mice clearing the infection at late timepoints (Fig. 3e, f, S2a, and S2b). In resolved livers, which were common at 21 dpi, we observed small clusters of various immune cells and small, flat areas of collagen; additionally, there were dead bacterial antigens still present (Fig. 3c, d). Such areas are consistent with a granuloma that has contracted and has almost completely resolved.

### Individual granulomas resolve at different rates

Bacterial burdens peak at 3 dpi and decrease over time thereafter (Fig. 3e and S2a). This decrease correlates with the appearance of the macrophage zone, suggesting that the switch of a microabscess to a granuloma halts bacterial replication. Starting at 7 dpi, some mice cleared the infection, and the proportion of mice that cleared steadily increased over time, although the kinetics of clearance vary slightly between experiments (Fig. 3e, 3f, S2a and S2b). The majority of mice cleared the infection between 14 and 35 dpi (Fig. 3f and S2a).

Infrequently at later time points, sporadic mice still had bacterial burdens (Fig. 3e and S2a). We hypothesize that these burdens arise from single, rare granulomas that have not yet resolved in the same organ where the vast majority of granulomas have sterilized the bacteria (Fig. 3a; arrow). In these rare granulomas, *C. violaceum* staining is still restricted to the necrotic core (Fig. S2c), which is more similar to a typical granuloma at 5 dpi. Therefore, it is most likely that all granulomas containing live bacteria restrict those bacteria in the core, whereas sterilized granulomas permit the trafficking of dead bacterial antigens to the macrophage zone. The total amount of liver inflammation (inflammatory regions, including granulomas at various phases of formation, maturation, and resolution) decreases over time with the visual disappearance of the necrotic core (Fig. 3g and S2d). This suggests that individual granulomas eliminate bacteria independently of each other within the same mouse. Indeed, when we dissected single granulomas from the same mouse at 7 dpi, some granulomas were sterile while others contained burdens, and a greater proportion became sterile at 10 dpi (Fig. S2e). These observations suggest that the granuloma response successfully clears *C. violaceum* infection.

We also made other histological observations described in Supplementary Notes and Fig. S3.

### Adaptive immunity is not required

Most, but not all, granuloma responses involve the adaptive immune system in which T cells maintain or support the organization of the macrophage zone[7,8,14,21,22]. However, in the *C. violaceum*-induced granuloma, T cells within the macrophage zone were sporadic and unorganized (Fig. 3d and S2f). Stochastic clearance of burdens began at 10 dpi in both WT and *Rag1*$^{-/-}$ mice (Fig. 3e, f, h, i). The proportion of mice that cleared the infection increased through 37 dpi when both WT and *Rag1*$^{-/-}$ had the same percentage of mice sterilizing the infection (Fig. 3j). The trend of clearance between WT and *Rag1*$^{-/-}$ mice is similar, which is consistent with previous results using a lower dose infection[3]. Furthermore, both groups equally survived the infection (Fig. 3k). The abundance and the architecture of granulomas per liver section was similar between WT and *Rag1*$^{-/-}$ at 3 dpi (Fig. 3l) and of three *Rag1*$^{-/-}$ mice analyzed by H&E staining at 26 dpi, two had no granulomas and one had a single small resolving granuloma (Fig. S2g). Although adaptive immunity is not required for this granuloma response, this does not preclude adaptive responses from organizing a more efficient immune defense during secondary infections. Taken together, *C. violaceum* induces the formation of innate granulomas that organize, eradicate the bacteria, and resolve without the need for adaptive immunity.

### Perforin and caspase-7 are not required for the granuloma response

Natural killer (NK) cell attack reduces *C. violaceum* burdens in the liver[3] through perforin and caspase-7[5]. We have shown cleaved caspase-7

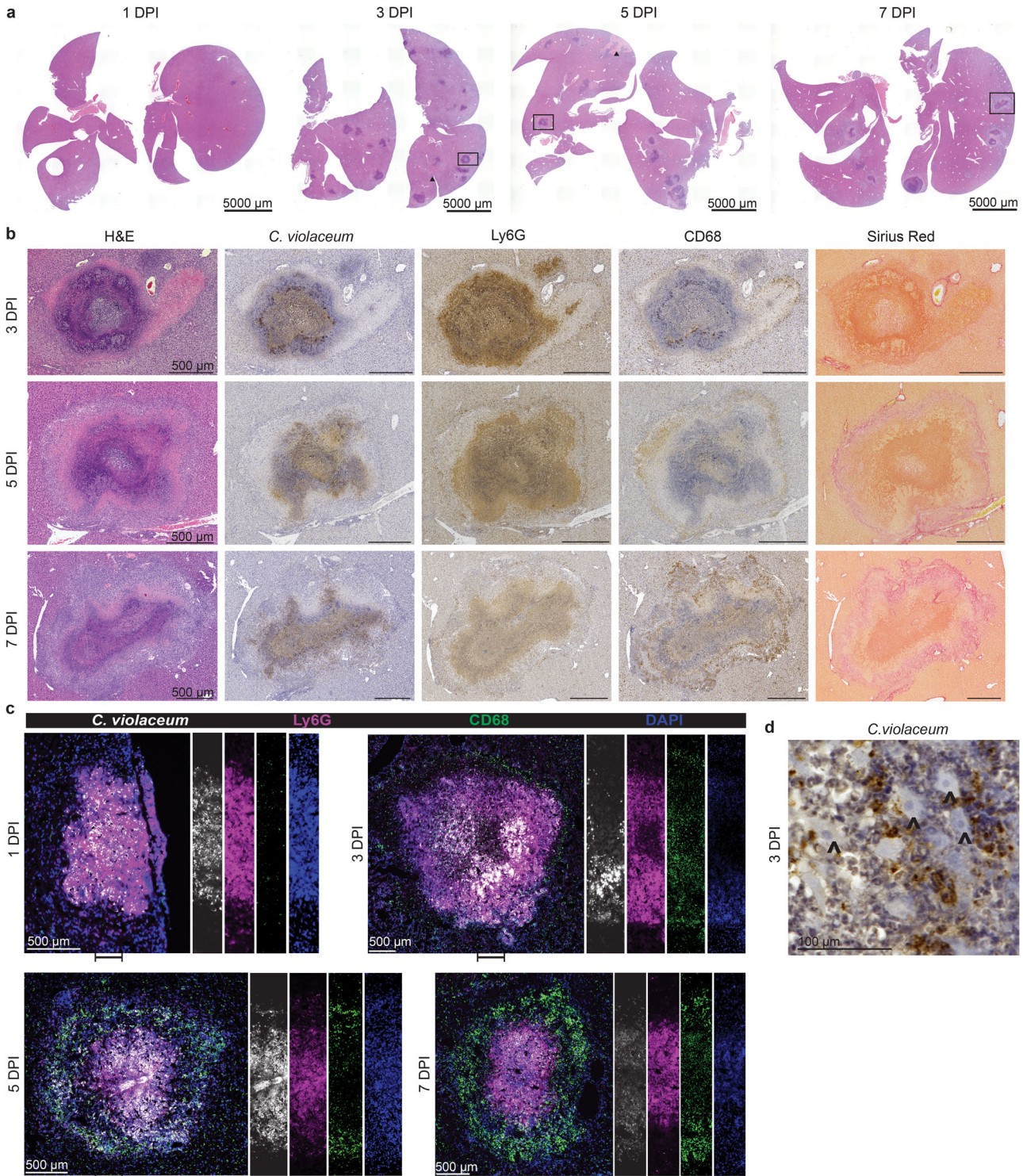

**Fig. 2 | A granuloma forms around the infected neutrophil swarm. a–d** Mice were infected with 10$^4$ CFU WT *C. violaceum*. **a** H&E staining of WT livers 1, 3, 5, and 7 dpi. Representative of 10 experiments (3 and 5 dpi) and 5 experiments (1 and 7 dpi), each with 3–4 mice, each with multiple granulomas per section. **b** Serial sections of WT liver stained by H&E or indicated IHC markers 3, 5, and 7 dpi. Representative of 10 experiments (3 and 5 dpi) and 5 experiments (7 dpi), each with 4 mice, each with multiple granulomas per section. **c** Serial sections of WT liver stained with indicated IF markers 1, 3, 5, and 7 dpi. Representative of 2 experiments, each with 4 mice, each with multiple granulomas per section. **d** Visualization of *C. violaceum* 3 dpi; carrot, coagulative necrotic hepatocytes. Representative of 10 experiments, each with 3–4 mice, each with multiple granulomas per section.

staining in what we now know are granulomas of WT mice[5], however, NK cells were scarce and unorganized within the macrophage zone (Fig. S3f; arrows). *Prf1*$^{-/-}$ mice survive and clear *C. violaceum* infection by 19 dpi (Fig. S3g and S3h), and both *Prf1*$^{-/-}$ and *Casp7*$^{-/-}$ mice have

similar granuloma architecture compared to WT mice (Fig. S3i). Thus, although some NK cells are present, neither perforin nor caspase-7 are required for the overall granuloma-mediated clearance of *C. violaceum*.

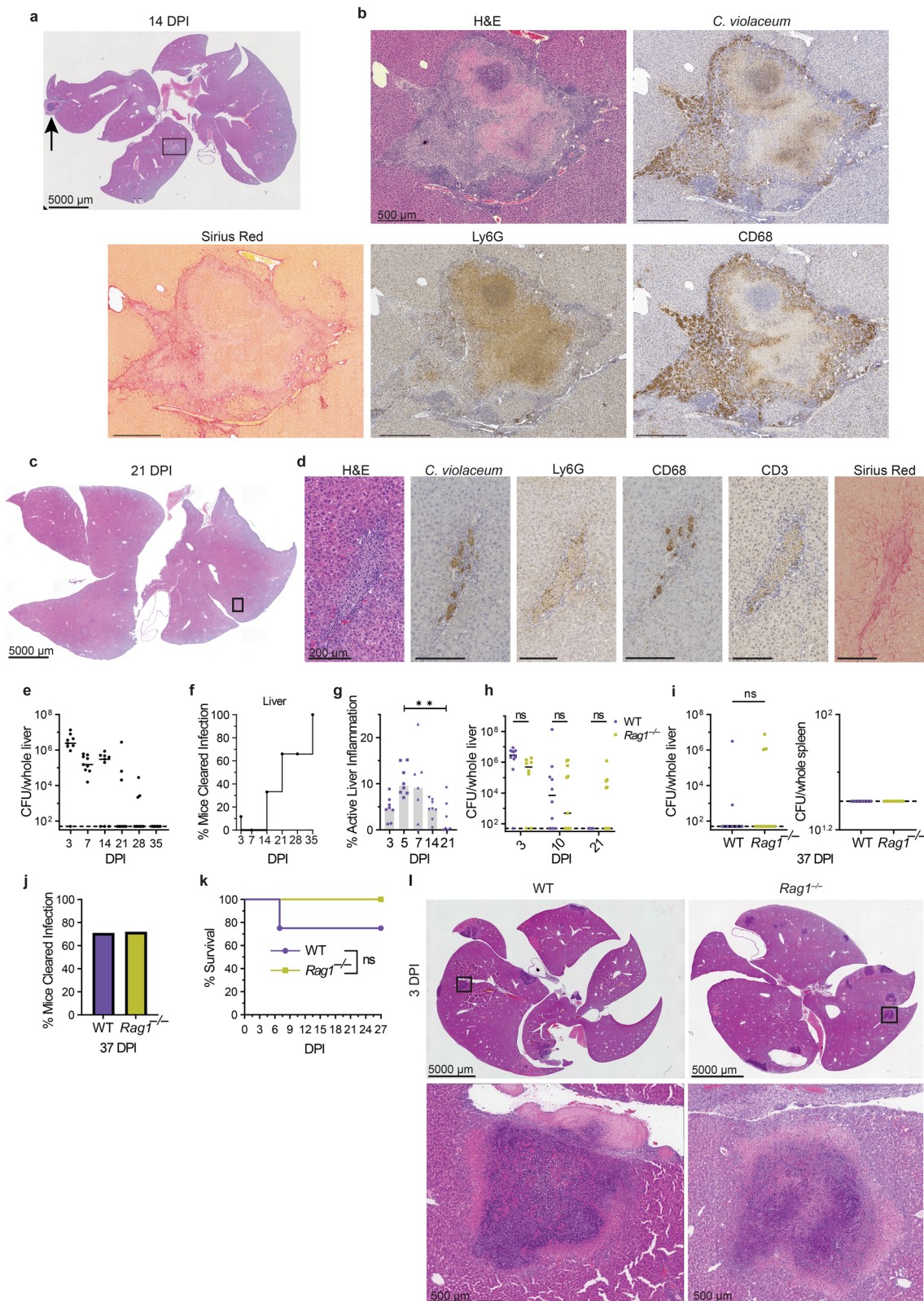

## Gasdermin D is essential during the granuloma response

*C. violaceum* uses a T3SS to reprogram host cells and the activity of this T3SS can be detected by the NAIP/NLRC4 inflammasome, activating caspase-1 which cleaves pro-IL-1β, pro-IL-18, and gasdermin D to their active forms. Once cleaved, gasdermin D oligomerizes and forms a pore that inserts into the inner leaflet of the plasma membrane, resulting in the influx of fluids causing the cell to swell and undergo

pyroptotic lysis. Pyroptosis is an inflammatory form of regulated cell death that can combat intracellular infection[30].

*Casp1/11*[−/−] and *Gsdmd*[−/−] mice infected with *C. violaceum* succumbed to infection between 3 and 9 dpi and had increased burdens in both the liver and spleen (Fig. 4a−c)[3,5]. We observed obvious abnormalities within the granuloma architecture of both knockouts at 3 dpi (Fig. 4d, e), with complete loss of the distinct layers seen in WT

**Fig. 3 | Adaptive immunity is not required to resolve the granuloma. a–l** Mice were infected with $10^4$ CFU WT *C. violaceum*. Dashed line, limit of detection; solid line, median. **a** H&E staining of WT liver 14 dpi. Representative of 3 experiments, each with 3–4 mice, each with multiple granulomas per section. Arrow, rare granuloma. **b** Serial sections of WT liver stained by H&E or indicated IHC markers 14 dpi. Representative of 3 experiments, each with 3–4 mice, each with multiple granulomas per section. **c** H&E staining of WT liver 21 dpi. Representative of 3 experiments, each with 3–4 mice, each with multiple granulomas per section. **d** Serial sections of WT liver stained by H&E or indicated IHC markers 21 dpi. Representative of 3 experiments, each 3–4 mice, each with multiple granulomas per section. **e**, **h**, and **i** Bacterial burdens in the liver and spleen at indicated timepoints. Representative of

2 experiments; each point is a single mouse. ns, not significant, by Two-way ANOVA (**h**) or Mann–Whitney test (**i**). **f** Bacterial burdens from **e** displayed as % bacterial clearance. **g** Percent active inflammation quantified per whole liver section stained with H&E. Data combined from 2 experiments; each point is a single mouse. \*\**p* = 0.0051, by Kruskal–Wallis test. **j** Bacterial burdens from (**i** liver) displayed as % bacterial clearance. **k** Survival analysis of WT and *Rag1*$^{-/-}$ mice. $N = 4$ (WT) or 7 (*Rag1*$^{-/-}$); Representative of two experiments. ns, not significant, by Kaplan–Meier survival analysis. **l** H&E staining of WT and *Rag1*$^{-/-}$ livers at 3 dpi. Representative of two experiments, each with 3, 4 mice per genotype, each with multiple granulomas per section.

mice (Fig. 4f, g). Most lesions in these knockout mice have increased abundance of necrosis containing Ly6G and *C. violaceum* staining (Fig. 4f, g). These lesions have a 'budding' morphology with strong Ly6G staining but little CD68 staining (Fig. 4f, g; H&E, brackets). When viewed in isolation, each single bud is reminiscent of a 1 dpi microabscess. Within each budding area and throughout each lesion in *Casp1/11*$^{-/-}$ and *Gsdmd*$^{-/-}$ mice, we observed *C. violaceum* staining that extended to the outer edge (Fig. 4f, g) and even outside the defective granulomas (Fig. S3j and S3k; arrows). This loss of bacterial containment could arise by escape from the defective granuloma or by seeding via portal circulation from the infected spleen. These data clearly demonstrate that caspase-1/11 and gasdermin D are required for the granuloma to form and contain *C. violaceum* infection.

## Spatial mapping of the granuloma transcriptome

To investigate gene expression within different zones of the granuloma, we used spatial transcriptomics (10x Genomics Visium) to capture mRNA from granulomas in WT mice during *C. violaceum* infection (Fig. 5a and S4a, Supplementary Data 1 and 2). Liver sections at 12 hpi, and 1, 3, 5, 10, 14, and 21 dpi contained granulomas that appear representative of each timepoint. We identified sixteen distinct expression clusters within the granuloma and the surrounding liver (Fig. 5a and S4a). Clusters had distinct distribution profiles within the layers of the granuloma architecture (Fig. 5a and S4b). To assign cell types within each capture area, we used published single-cell sequencing data from the Mouse Cell Atlas 1.0 (https://doi.org/10.1016/j.cell.2018.02.001; Fig. 5b, 5c, S4b).

Immune cell-enriched *Ptprc* (CD45)-positive areas were located on the left half of the UMAP plot, while non-immune clusters were on the right and generally expressed the hepatocyte marker *Alb* (Fig. S4c and S4d). We annotated clusters by both sequencing- and histological-based parameters: necrotic core (NC)—identified based on basophilic properties and low RNA abundance; coagulative necrotic zone (CN)—identified based on spatial overlap with regions of diffuse tissue damage surrounding the necrotic core; CN-macrophage zone (CN-M)—identified based on presence at the nexus between the CN and M zones; macrophage zone (M)—identified based on the mildly basophilic nature of the underlying tissue and presence outside the CN zone (Figs. 5a, 5c and S4a). Sub-zonal annotation was used to denote unique clusters found in the same zone. We created gene expression modules using markers from published single-cell RNA sequencing datasets from mouse liver[31,32] and cross-referenced these markers in ImmGen for cell type-specific expression, and selected marker genes expressed in our spatial transcriptomics dataset. We created modules for neutrophils, monocytes, macrophages, T cells, fibroblasts, hepatocytes, and endothelial cells (Fig. S4e). The neutrophil module peaked at 3 dpi, after which their expression decreased but remained present over time (Fig. 5d). Monocyte prevalence peaked at 3 dpi and declined thereafter (Fig. 5d), suggesting that the monocytes differentiated into macrophages, which also peaked at 3 dpi but remained steady throughout the granuloma response (Fig. 5d). Consistent with their lack of involvement in the granuloma response, T-cell gene expression

was minimal (Fig. 5d). Fibroblast gene expression peaked at 5 dpi and remained steady until later timepoints (Fig. 5d).

By assigning modules to different zones within the granuloma, we observed that neutrophil gene expression was present primarily within the necrotic core and the coagulative necrotic layers (Fig. 5e), which histologically are composed of necrotic debris, indicating that mRNAs from dead neutrophils persist. Macrophage and monocyte gene expressions were present primarily throughout the coagulative necrotic zone, macrophage zone, and just outside the macrophage zone (Fig. 5e) but were minimal in the necrotic core. Fibroblast gene expression was seen throughout the macrophage zone spreading into the coagulative necrotic zone (Fig. 5e), consistent with Sirius Red staining (Fig. 2b). Hepatocyte and endothelial cell gene expression was diminished within the granuloma and was predominant in the outer surrounding tissue (Fig. 5e). Using key genes from each cell module, we confirmed their location within the granuloma by visualizing spatial expression in each capture area (Fig. 5f). Collectively, these spatial transcriptomics data confirm the histologically visualized distribution of cell types and the specific staining for cell types with marker-specific antibodies.

## Nitric oxide is required to clear *C. violaceum* in the granuloma

*Nos2* and *Acod1* have both been previously implicated in other granuloma models[33–37] and we observed expression of both in our spatial transcriptomics data. *Nos2* encodes inducible nitric oxide synthase (iNOS), an enzyme that produces nitric oxide (NO) that can kill bacteria. *Acod1* encodes IRG1, an enzyme that produces itaconate, which can also kill bacteria and has immunometabolic activities[38,39]. Indeed, we observed that NO can kill *C. violaceum* in vitro, and although itaconate alone was not toxic, it enhanced NO toxicity (Fig. S5a). This suggested that iNOS and IRG1 might cooperate to sterilize the granuloma. However, *Acod1*$^{-/-}$ mice survived *C. violaceum* infection with normal bacterial burdens (Fig. S5B and S5c). Therefore, although IRG1 is expressed in the granuloma, it is not essential for clearance of *C. violaceum*.

In contrast, all *Nos2*$^{-/-}$ mice succumbed to *C. violaceum* infection (Fig. 6a), confirmed with littermate controls (Fig. S6a). The importance of iNOS was not yet manifested at 3 dpi, when *C. violaceum* continued to replicate normally in the liver and continued to be cleared from the spleen of *Nos2*$^{-/-}$ mice (Fig. 6b), suggesting that iNOS did not act during the neutrophil swarm phase of the infection. Bacterial burdens increased in the liver of *Nos2*$^{-/-}$ mice at 5 dpi and continued to increase at 7 dpi compared to WT mice, and *Nos2*$^{-/-}$ mice succumbed shortly thereafter (Fig. 6a–c). The spleen also showed high *C. violaceum* burdens for the majority of the *Nos2*$^{-/-}$ mice at these later time points (Fig. 6b, c), which may reflect dissemination from the liver rather than an open niche in the spleen.

*Nos2*$^{-/-}$ mice had greater affected areas of the liver, and their granulomas were markedly abnormal (Fig. 6d). Granulomas in *Nos2*$^{-/-}$ mice did have a necrotic core surrounded by a coagulative necrotic zone. However, in contrast to WT mice, beyond the coagulative necrotic hepatocytes we observed multiple zones of Ly6G-positive necrosis with additional fragmented coagulative necrotic zones interspersed. *C. violaceum* staining appears lighter compared to WT

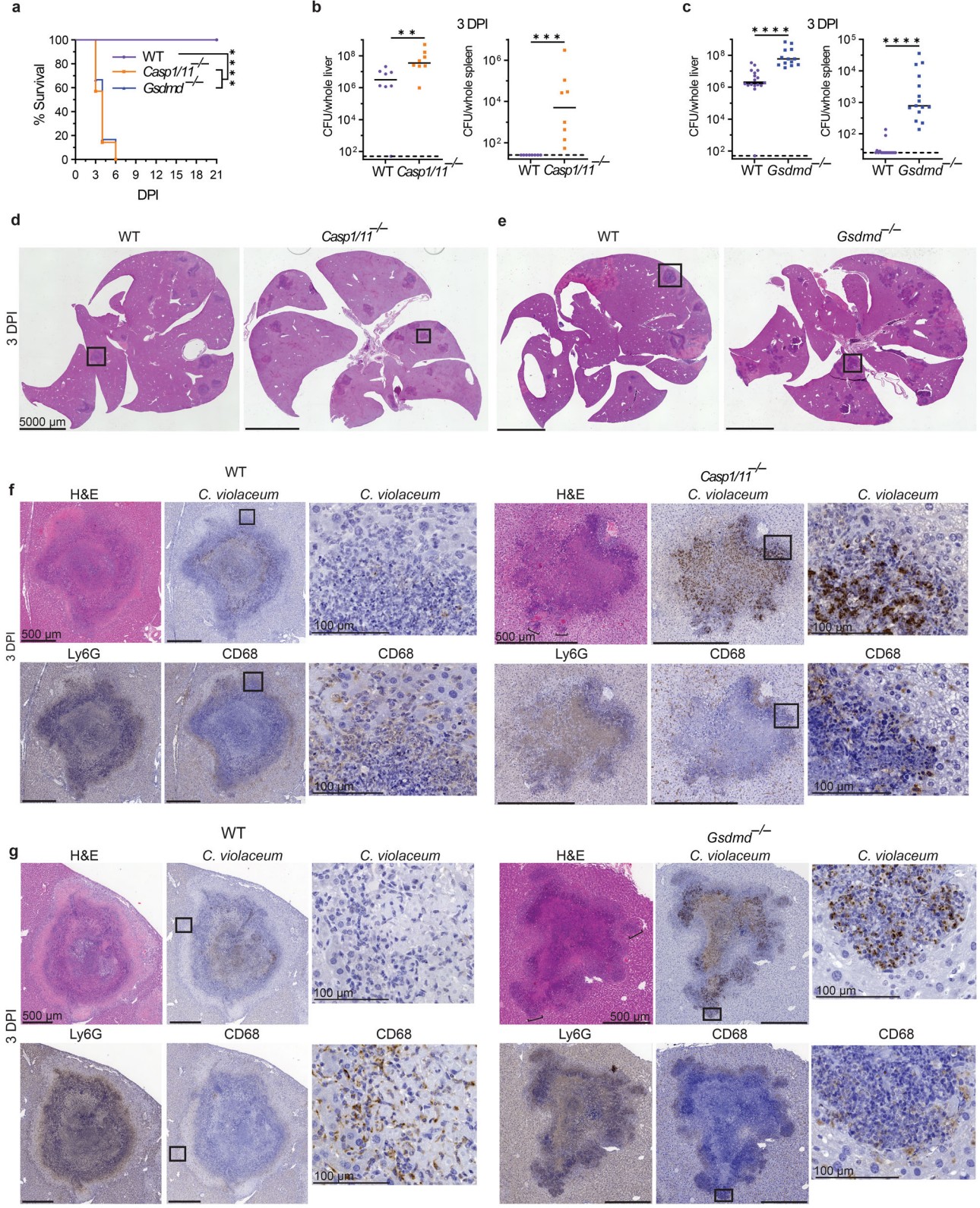

**Fig. 4 | Gasdermin D is essential during the granuloma response. a–g** Mice were infected with $10^4$ CFU WT *C. violaceum*. Dashed line, limit of detection; solid line, median. **a** Survival analysis of WT, *Casp1*[-/-] *Casp11*[-/-] (*Casp1/11*[-/-]), and *Gsdmd*[-/-] mice. $N = 8$ (WT), 7 (*Casp1/11*[-/-]), or 6 (*Gsdmd*[-/-]) mice. Representative of two experiments, each with 6–8 mice per genotype. ****$p < 0.0001$ (WT vs *Casp1/11*[-/-]), ****$p = 0.0001$ (WT vs *Gsdmd*[-/-]), or n.s. $p = 0.7837$ (*Casp1/11*[-/-] vs *Gsdmd*[-/-]), by Kaplan–Meier survival analysis with Bonferroni correction for multiple comparisons ($\alpha = 0.0167$).

**b, c** Bacterial burdens in the liver and spleen at 3 dpi. Data combined from two experiments; each point is a single mouse. **$p = 0.0070$, ***$p = 0.0002$, ****$p < 0.0001$, by Mann–Whitney test. **d–g** Serial sections of WT, *Casp1*[-/-]*Casp11*[-/-] (*Casp1/11*[-/-]), and *Gsdmd*[-/-] livers stained by H&E or indicated IHC markers 3 dpi. Representative of two experiments, each with 3–4 mice per genotype, each with multiple granulomas per section.

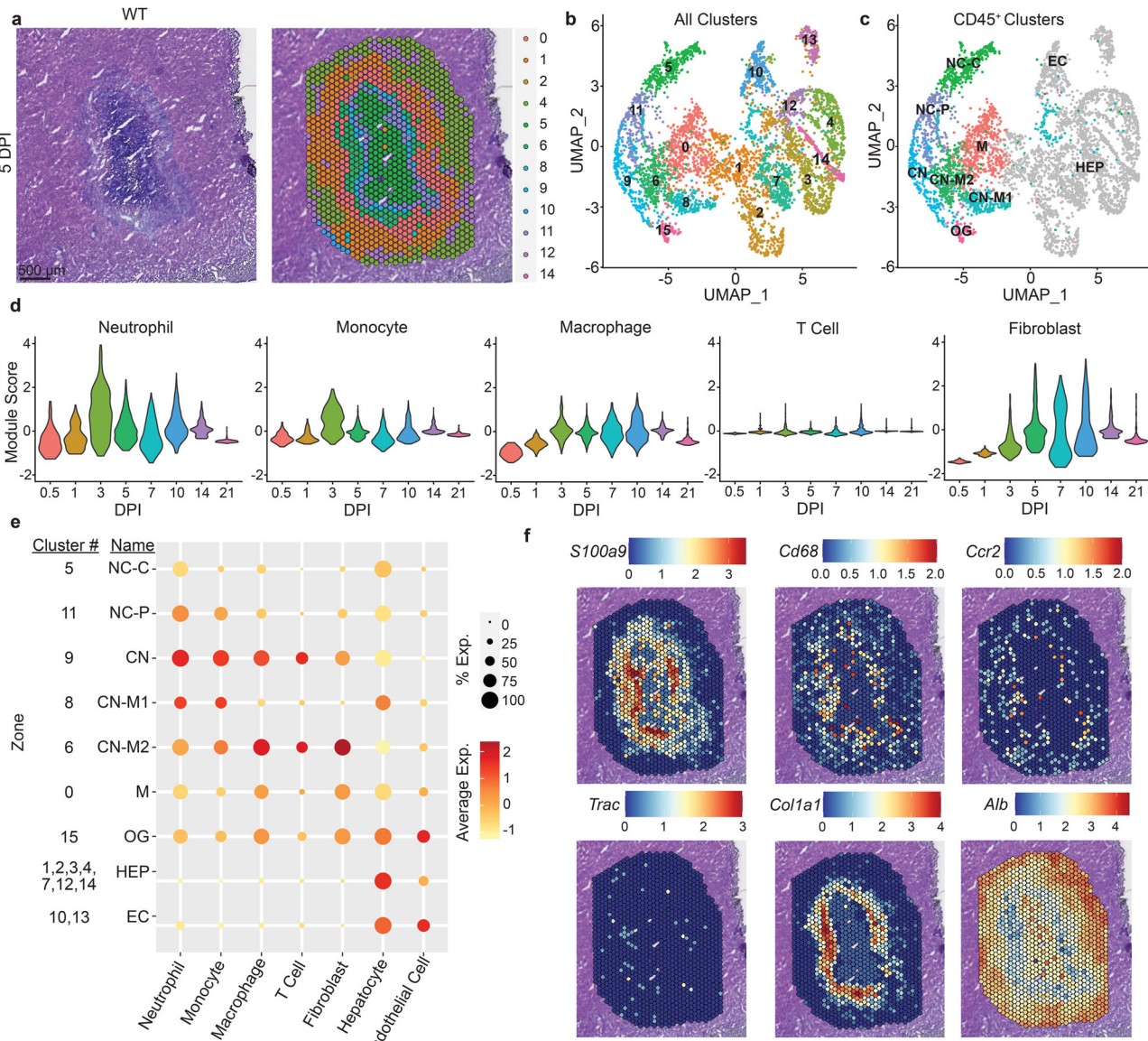

**Fig. 5 | Spatial mapping of the granuloma transcriptome. a–f** Mice were infected with 10⁴ CFU WT *C. violaceum* and one liver was harvested per indicated timepoint for spatial transcriptomic analysis. **a** H&E and spatial transcriptomic orientation of indicated clusters from same granuloma at 5 dpi. Spatial transcriptomics was only completed on one experiment. **b** UMAP plot for cluster expression orientation for all clusters from all timepoints of infection. **c** UMAP plot of cluster expression in designated locations within granuloma architecture. Necrotic core (NC), necrotic core center (NC-C), necrotic core periphery (NC-P), coagulative necrotic zone (CN), CN-macrophage zone (CN-M), macrophage zone (M), outside granuloma (OG), hepatocyte (HEP), and endothelial cell (EC). **d** Module scores of various immune cells and fibroblast expression markers from granulomas throughout infection time course. **e** Indicated cell type expression dot plot in each cluster within the granuloma architecture; combined from all timepoints of infection. **f** Spatial transcriptomic expression of indicated genes representing immune and non-immune cells within 5 dpi granuloma. *S100a9* (neutrophils), *Cd68* (macrophages), *Ccr2* (monocytes), *Trac* (T cells), *Col1a1* (fibroblasts), and *Alb* (hepatocytes and endothelial cells).

granulomas, however, closer examination revealed that the bacteria were more dispersed throughout the granuloma and present in more discrete clusters akin to a 1 dpi lesion in WT mice (Figs. 6d and 1j). This suggests that *C. violaceum* breached the granuloma and initiated new neutrophil swarms. Indeed, we observed *C. violaceum* staining throughout the zones in *Nos2⁻/⁻* mice, even reaching the edge of the granuloma.

Abnormalities in the macrophage zone in *Nos2⁻/⁻* mice varied between granulomas. Some *Nos2⁻/⁻* granulomas had single breaches in the macrophage zone (shown below in Fig. 7g), whereas other granulomas lost the macrophage zone altogether (Fig. 6d). This loss of architecture could be due to the large influx of neutrophils that have overrun the granuloma response, potentially disrupting nascent

macrophage zones. Neutrophil and macrophage numbers in the liver were equal by flow cytometry at 3 dpi in WT and *Nos2⁻/⁻* mice, but at 7 dpi we observed significantly more neutrophils and fewer macrophages in *Nos2⁻/⁻* mice (Fig. 6e and S5d). This is consistent with histological staining for neutrophil and macrophage markers (Fig. 6d).

We used spatial transcriptomics to visualize the general location of *Nos2* expression within the granuloma architecture of WT mice over time (Fig. 6f). As early as 12 hpi we observed modest *Nos2* expression, which became strongest at 3 dpi and continued throughout the infection (Fig. 6f). This expression pattern correlated with elevated bacterial burdens in *Nos2⁻/⁻* mice but was offset by two days (Fig. 6b). At 3 dpi, macrophages are maximizing *Nos2* mRNA, but bacterial burdens in *Nos2⁻/⁻* mice are not elevated until 5 dpi (Fig. 6b). To

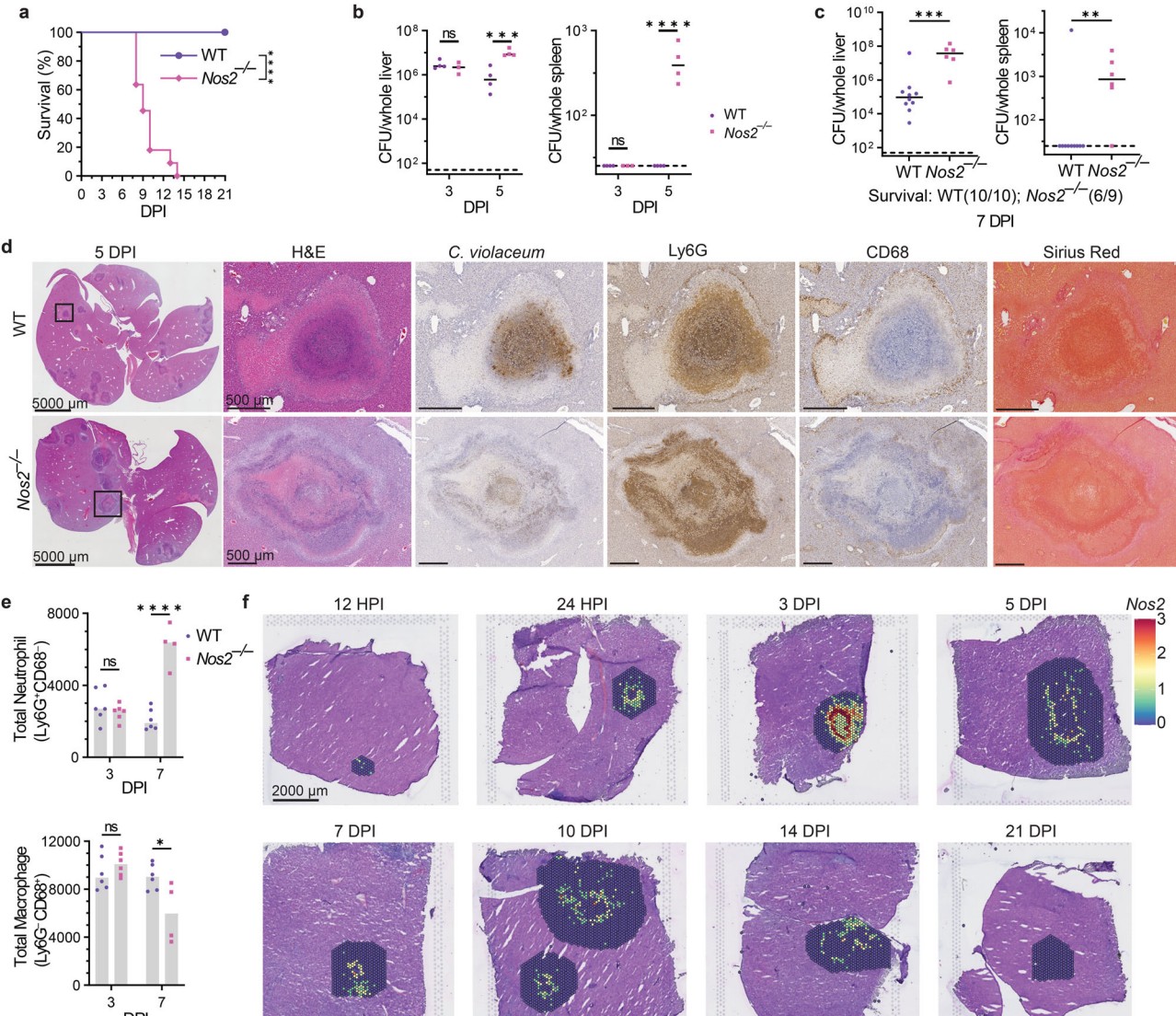

**Fig. 6 | Nitric oxide is required to clear *C. violaceum* in the granuloma. a–f** Mice were infected with $10^4$ CFU WT *C. violaceum*. Dashed line, limit of detection; solid line, median. **a** Survival analysis of WT and *Nos2*[-/-] mice. Representative of 2 experiments, each with 11 mice per genotype. ****$p < 0.0001$, by Kaplan–Meier survival analysis. **b** Bacterial burdens in the liver and spleen at 3 and 5 dpi. Representative of 2 experiments; each point is a single mouse, 3–4 mice per day per genotype. ns, not significant, ***$p = 0.0007$, ****$p < 0.0001$, by two-way ANOVA. **c** Bacterial burdens in the liver and spleen at 7 dpi. Combined 2 experiments, each point is a single mouse. $N = 10$ (WT) or 9 (*Nos2*[-/-]) mice. **$p = 0.0076$ by

Mann–Whitney and ***$p = 0.0005$ by unpaired two-tailed $t$ test. **d** Serial sections of WT and *Nos2*[-/-] livers stained by H&E or indicated IHC markers 5 dpi. Representative of two experiments, each with 3–4 mice per genotype, each with multiple granulomas per section. **e** Flow cytometry of total neutrophil and total macrophage numbers from WT and *Nos2*[-/-] livers 3 and 7 dpi. Data combined from two experiments. ns, not significant, *$p = 0.0114$, and ****$p < 0.0001$, by two-way ANOVA. **f** Spatial transcription of *Nos2* expression within WT granulomas at indicated timepoints. One liver was harvested per the indicated timepoint for spatial transcriptomic analysis.

reconcile this discrepancy, we stained for iNOS protein. Despite the detection of *Nos2* mRNA as early as 12 and 24 hpi, iNOS protein staining was not detectable at 1 dpi, and staining at 3 dpi was quite faint (Fig. S6b). iNOS staining became prominent only at 5 dpi (Fig. S6c). Thus, *Nos2* mRNA expression precedes detectable iNOS protein content as well as bacterial clearance. Altogether, these data demonstrate that granuloma macrophages express iNOS, that NO can kill *C. violaceum*, and in the absence of iNOS the granuloma cannot maintain its organized architecture.

### Gasdermin D and iNOS are independently required to maintain the granuloma architecture

Both *Gsdmd*[-/-] and *Nos2*[-/-] mice have disrupted granuloma architecture and fail to clear *C. violaceum*. We examined whether iNOS is downstream of gasdermin D in this granuloma response. This was not

the case, since *Nos2* expression and NO production in *Gsdmd*[-/-] mice were elevated compared to WT mice (Fig. 7a, b). iNOS has been implicated in the regulation of IL-1β in other infectious models[34], however, during *C. violaceum* infection, *Il1b*[-/-] mice form normal granulomas and survive the infection (Fig. S6d and S6e). Furthermore, *Nos2*[-/-] macrophages remain competent to undergo pyroptosis after *C. violaceum* infection (Fig. S6f and S6g)[3]. Furthermore, burdens per granuloma were elevated in *Casp1/11*[-/-] and *Gsdmd*[-/-] mice at 3 dpi, but not in *Nos2*[-/-] mice (Fig. 7c). Whereas *Casp1/11*[-/-] and *Gsdmd*[-/-] mice support splenic *C. violaceum* replication at 3 dpi, *Nos2*[-/-] have little to no spleen burdens at this timepoint (Figs. 4b, 4c, and 6b). To quantitate the defect in the overall granuloma response, we measured the total area of active inflammation in the liver as a percentage of the total tissue area. This included functional granulomas in WT mice, defective granulomas in the various knockout

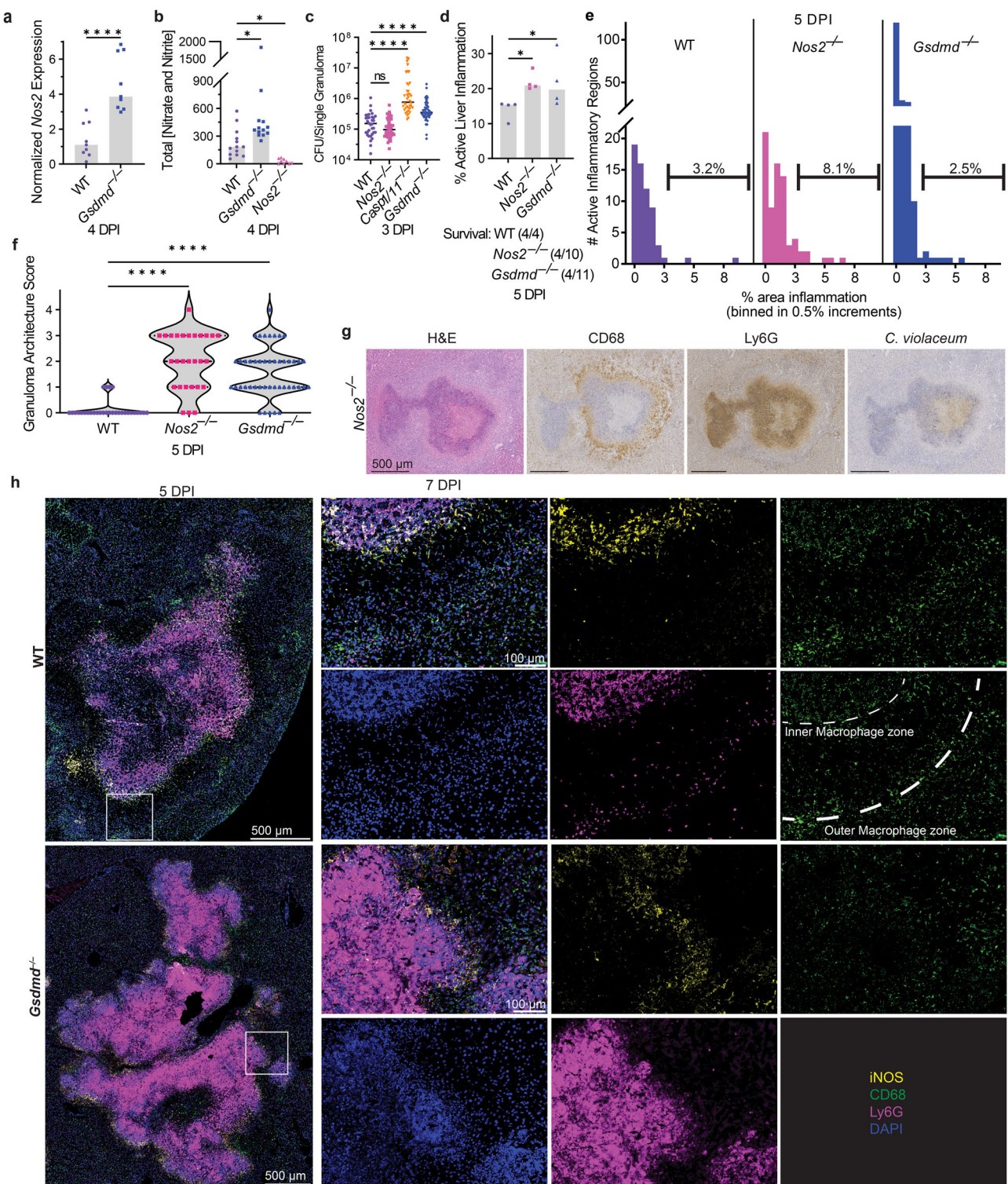

mice, as well as newly seeded microabscesses seen in the *Casp1/11⁻ᐟ⁻* and *Gsdmd⁻ᐟ⁻* mice. Both *Casp1/11⁻ᐟ⁻* and *Gsdmd⁻ᐟ⁻* mice have increased inflammation as well as increased numbers of inflammatory regions within the liver at 3 dpi (Fig. S6h, S6i, S6j, and S6k), which correlates with increased bacterial burdens (Fig. 4b, c). *Gsdmd⁻ᐟ⁻* mice have increased inflammation as well as a higher number of small inflammatory regions compared to WT at 5 dpi, akin to 1 dpi microabscesses (Fig. 7d, e). These data suggest that pyroptosis-deficient mice have a double failure during *C. violaceum* infection. First, the 'budding' morphology indicates a failure of the granuloma to contain *C. violaceum* from the local tissue. Second, the

small microabscesses may be due to bacterial dissemination from the spleen, first seen at 3 dpi (Fig. S3j and S3k).

In contrast, *Nos2⁻ᐟ⁻* mice did not have increased inflammation, inflammatory regions, or burdens compared to WT mice at the same 3 dpi timepoint (Figs. S6h, S6i, S6j, S6k and 6b). It was not until 5 dpi that the surviving *Nos2⁻ᐟ⁻* mice have increased burdens, as well as increased inflammation (to a similar degree as those *Gsdmd⁻ᐟ⁻* mice that survived until this timepoint) (Figs. 6b and 7d). There was selection bias for the knockout mice that survived until the 5dpi timepoint, but we hypothesize that the mice that did not survive to this timepoint had worse pathology (Fig. 7d). When we quantitated the inflammatory regions at

**Fig. 7 | Gasdermin D and iNOS are independently required to maintain the granuloma architecture. a–h** Mice were infected with $10^4$ CFU WT *C. violaceum*. **a** qPCR of *Nos2* gene expression in WT and *Gsdmd*$^{-/-}$ granulomas 4 dpi. Representative of two experiments; each point is 5–6 pooled granulomas harvested from single mouse. ****$p < 0.0001$, by Mann–Whitney test. **b** Total nitrate and nitrite concentrations in serum from WT, *Gsdmd*$^{-/-}$, and *Nos2*$^{-/-}$ mice 4 dpi. Representative of two experiments performed in triplicate. *$p = 0.0471$ (WT vs *Gsdmd*$^{-/-}$) or $p = 0.0167$ (WT vs *Nos2*$^{-/-}$), by Kruskal–Wallis test. **c** Bacterial burdens of single granulomas 3 dpi. Data combined from two experiments; each point is a single granuloma. Solid line, median. n.s. nonsignificant, ****$p < 0.0001$ by One-way ANOVA test. **d, e** Percent active inflammation quantified per whole liver section stained with H&E 5 dpi. Representative of two experiments, each with four mice per genotype. **d** Total percent active liver inflammation; each point is a single mouse. *$p = 0.0482$ (WT vs *Gsdmd*$^{-/-}$) or $p = 0.0285$ (WT vs *Nos2*$^{-/-}$), by Kruskal–Wallis test. Text below the graph denotes mice that died prior to harvest; these mice were not included in the histology analysis. **e** Percent active liver inflammation per each individual inflammatory region from **d**. Data combined from all mice within genotype from **d**. Frequency distribution histogram by the size of inflammatory region, binned in 0.5% increments. Brackets indicate total percentage of inflammatory regions that are above 3% area in size. **f** Granulomas scored for failure of granuloma architecture 5 dpi. Zero indicates fully intact granuloma architecture; intermediate score includes granulomas with discreet breaches in the macrophage zone; four represents a complete loss of architecture. Data combined from 2 experiments, each with 3, 4 mice per genotype, each with multiple granulomas per section. ****$p < 0.0001$, by Kruskal–Wallis test. **g** Serial sections of *Nos2*$^{-/-}$ liver stained by H&E or indicated IHC markers showing a failed granuloma 7 dpi. Representative of 2 experiments, each 3, 4 mice, each with multiple granulomas per section. **h** WT and *Gsdmd*$^{-/-}$ liver sections stained with indicated IF markers 5 dpi. Representative of 2 experiments, each with 4 mice per genotype, each with multiple granulomas per section.

5 dpi, we saw that *Nos2*$^{-/-}$ mice had a similar number of inflammatory regions as WT (Fig. S6l), but these regions are larger with the median region size greater than WT (Fig. 7e; brackets). This results in a greater total area of inflammation (Fig. 7d). Taken together, these data suggest that mice deficient in iNOS have a granuloma defect that allows local spread in the liver.

Defective granulomas are heterogeneous. To quantitate the severity of failed granuloma architecture, a board-certified pathologist blindly scored each granuloma for the presence of granuloma zones, containment of Ly6G-positive cells, and the integrity of the macrophage zone (Fig. 7f). A score of zero indicates fully intact granuloma architecture (as in Fig. 1c). An intermediate score includes granulomas with discreet breaches in the macrophage zone. Such breaches are typically connected to a budding morphology that consists of a neutrophil swarm with *C. violaceum* staining that appears similar to a 1 dpi lesion (Fig. 7g compared to Fig. 1j). A score of four represents a complete loss of architecture (e.g., *Gsdmd*$^{-/-}$ in Fig. 4g and *Nos2*$^{-/-}$ in Fig. 6e). Both *Gsdmd*$^{-/-}$ and *Nos2*$^{-/-}$ mice had a markedly increased number of failed granulomas compared to WT mice (Fig. 7f). These data indicate gasdermin D and iNOS are both independently required to maintain the granuloma architecture.

We further examined *Nos2* expression in the granuloma using the spatial transcriptomic dataset. *Nos2* expression was associated with clusters that contain CD45-expressing immune cells (Fig. S6m and 5c); the highest levels of *Nos2* expression were in clusters 6, 8, 9, and 11 (Fig. S6m). These clusters with high *Nos2* expression were located at the periphery of the necrotic core and within the coagulative necrotic zone, and surprisingly less associated with the macrophage zone (Fig. 5e). We had previously only noted a single macrophage zone at the periphery of the granuloma. However, instructed by these *Nos2* spatial expression patterns, we reevaluated the granuloma histology focusing on the border between the necrotic core and the coagulative necrotic hepatocytes. Indeed, in addition to the "peripheral macrophage zone" we observed a second, albeit thinner, "inner macrophage zone" between the coagulative necrotic hepatocytes and the necrotic core (Fig. 7h and 2c). Therefore, in addition to validating gene candidates, spatial genomics highlighted new granuloma zones that were less apparent by H&E staining.

We next stained for iNOS protein to determine the relative expression in these two macrophage zones. Consistent with the spatial transcriptomics data, iNOS overlapped with CD68 staining in the inner macrophage zone (Fig. 7h and S6c). *Gsdmd*$^{-/-}$ mice have lost their granuloma architecture but retain CD68 staining, albeit discontinuously, at the periphery of the granuloma. Concomitantly, iNOS staining was maintained at the periphery of the granuloma and was also weakly visible within hepatocytes identified by nuclear staining pattern and lack of CD68 and Ly6G staining (Fig. 7h and S6c). Hepatocytes are known to produce NO via iNOS under stressful conditions[40,41], which could occur due to the failed granuloma response in *Gsdmd*$^{-/-}$ mice.

However, the NO produced by hepatocytes is not enough to rescue the failed granuloma response in these knockout mice. We again observed that *Gsdmd*$^{-/-}$ and *Nos2*$^{-/-}$ mice had *C. violaceum* staining throughout the entire granuloma, not limited to the core (Fig. S6n). Therefore, both gasdermin D and iNOS are required to keep *C. violaceum* contained in the granuloma core. The *C. violaceum*-induced granuloma requires at least two separate defense pathways, gasdermin D and iNOS, to maintain the integrity of the granuloma architecture, which is essential to eradicate *C. violaceum* infection.

## Discussion

*C. violaceum* first infects hepatocytes and rapidly leads to a neutrophil swarm within 1 day. Remarkably, *C. violaceum* continues to replicate through day 3 despite the copious numbers of neutrophils. This is a catastrophic failure of the innate immune response. *C. violaceum* has numerous virulence traits, including T3SS effectors (most of whose functions remain unknown[2]) which could cause this neutrophil failure. In other infectious models, such as *Yersinia* species, the same neutrophil failure could arise from virulence traits that mitigate neutrophil function[42]. Indeed, pyogranulomas form in response to *Yersinia pseudotuberculosis* infection in the small intestine, which is driven by monocytes[12,43–46]. We speculate that during *C. violaceum* and *Y. pseudotuberculosis* infections it is the failure of neutrophils that triggers the granuloma response. Neutrophils have also been implicated in the granuloma response to *M. tuberculosis*, with recent studies showing that *M. tuberculosis* preferentially occupies and replicates in live or necrotic neutrophils[47–49]. However, this occurs once the granuloma is already formed, and other studies show neutrophils are not the first immune cell to be infected[50,51]. Thus, whether neutrophils play a key role in granuloma initiation remains unclear in the *M. tuberculosis*-induced granuloma. In contrast, by providing well-defined stages and investigating early events that precede granuloma formation, we propose the *C. violaceum*-induced granuloma may be revealing a fundamental reason that would explain granuloma formation.

In studies of *Mycobacterium tuberculosis*, infections in pyroptosis-deficient mice sometimes have exacerbated phenotypes and a failed granuloma response[52], whereas other studies show no role[53,54]. This contrasts with the clear and robust protective role of pyroptotic proteins in the *C. violaceum*-induced granuloma. Although *C. violaceum* can inhibit the apoptotic caspases[6,55], it does not inhibit the pyroptotic caspases. We demonstrated that caspase-1/11 and gasdermin D-mediated defenses are essential to maintain granuloma architecture. Without these pyroptotic proteins, mice still express *Nos2* and generate NO, but this antibacterial mechanism fails to clear the infection. During *C. violaceum* infection, WT mice have less total liver inflammation from the granuloma response compared to *Casp1/11*$^{-/-}$ and *Gsdmd*$^{-/-}$ mice, and these pyroptotic proteins prevent the spread of the bacteria to the edge of the granuloma. We speculate that pyroptosis prevents *C. violaceum* from establishing a replicative

niche within the organized granuloma macrophage zones. Thus, we do not see a role for the pyroptotic proteins in driving a pathological exacerbation of the inflammatory response, and instead we demonstrate a clearly beneficial role of pyroptosis during the successful granuloma response.

Many other granuloma models have had contrasting conclusions as to whether caspase-1/11 and gasdermin D contributed to granuloma biology. During *Schistosomiasis*, where granuloma formation is pathological, multiple pyroptotic pathway knockout mice have smaller granulomas[56–58], suggesting that pyroptotic proteins are detrimental. This conclusion is the opposite to our conclusion with *C. violaceum* and may be due to the fact that schistosomes are large extracellular parasites, thus pyroptosis is not needed to prevent intracellular infection. Similarly, in a noninfectious granuloma response, *Nlrp3*[−/−] mice treated with trehalose 6.6′-dimycolate, a mycobacterial glycolipid, had fewer granulomas compared to WT mice[59]. Thus, in models where granulomas drive pathological tissue damage, inflammasomes can exacerbate pathology.

We show that iNOS is absolutely essential for the granuloma response to *C. violaceum*. WT mice use iNOS to eradicate *C. violaceum* and therefore survive; in contrast, *Nos2*[−/−] mice succumb to the infection due to granuloma failure. *Nos2*[−/−] mice also have increased susceptibility to *M. tuberculosis* infection[47], however, unlike *C. violaceum* infection, iNOS fails to clear *M. tuberculosis* in WT mice[60–63]. We speculate that *M. tuberculosis* has strong NO resistance mechanisms, thwarts the sterilizing effects of NO, and thereby is able to survive in granulomas long term[60]. During *M. tuberculosis* infection, NO also plays an immunomodulatory role by suppressing IL-1 production to limit pathology and prevent neutrophil recruitment[34,35]. If we are correct that *M. tuberculosis* is highly resistant to NO, it may be that the primary role of NO during *M. tuberculosis* infection is to modulate the inflammatory response. In our model, we also observe increased neutrophil recruitment and a loss of granuloma architecture in *Nos2*[−/−] mice, however, if *C. violaceum* is sensitive to NO, this cannot be separated from immunomodulation. We speculate that during the *C. violaceum*-induced granuloma response, the primary role of NO is to kill bacteria, and impeding neutrophil influx is a secondary function that may or may not be essential. There is another key difference of iNOS function between these two granuloma-inducing pathogens—during *M. tuberculosis* infection, *Nos2* expression is driven by the adaptive immune response[34], whereas during *C. violaceum* infection the adaptive immune response is dispensable. Thus, the *C. violaceum*-induced granuloma demonstrates that innate immunity can use iNOS to sterilize a granuloma without T-cell help.

The current understanding of granuloma biology remains perplexing, despite being studied for over a century in various granuloma models[7,14]. Granuloma biology is intriguing due to the sophisticated organization of immune cells unified into a multicellular collective to combat a pathogen that poses a dire threat to the host. Understanding how a granuloma operates requires animal model systems where the complexities of the immune response can be studied in their entirety, from the very first hours of infection to the final stages of resolution. *C. violaceum* provides such a model, where a granuloma rapidly forms and efficiently eradicates this environmental pathogen (Fig. 8). That the granuloma successfully clears *C. violaceum* is remarkable because many pathogens survive within granulomas chronically. It may be that such host-adapted pathogens have virulence factors that thwart the efficacy of the granuloma, whereas environmental pathogens such as *C. violaceum* lack this capacity. Therefore, the *C. violaceum*-induced granuloma model could be a new way to understand the complexities of granuloma biology and clearly demonstrates that granulomas are potent immune defenses that can successfully eradicate infection.

## Methods

### Ethics statement
Animal protocols were approved by the Institutional Animal Care and Use Committee (IACUC) at the University of North Carolina at Chapel Hill or by the IACUC at Duke University and met guidelines of the US National Institutes of Health for the humane care of animals.

### Bacterial strains and culture conditions
*Chromobacterium violaceum* ATCC 12472 was used in all experiments except mixed inocula where the nalidixic acid spontaneous resistant mutant CVN was used as WT and was mixed with CVN Δ*vioA* strain at 1:1 for infection. All strains were grown on brain heart infusion (BHI) agar plates overnight at 37 °C. Bacteria were subcultured in 2 mL BHI broth with aeration at 37 °C overnight and directly diluted to $1 \times 10^4$ CFU/mL in PBS for infection inoculum.

### Mouse infection and survival
For all experiments, 8- to 12-week-old mice were infected via intraperitoneal route at indicated CFUs of bacteria in PBS. Mice were monitored for survival or whole livers, spleens, and individual granulomas were harvested at indicated timepoints, homogenized, and plated on BHI agar as described above.

### Mouse strains
All mouse strains were bred and housed at Duke University in a specific pathogen-free facility. For infection, mice were transferred to a BSL2 infection facility within Duke University, and mice were allowed to acclimate for at least two days before infection. Wild-type C57BL/6 (referred to as WT; Jackson Laboratory), *Ncf1*[mt/mt] (referred to as *Ncf1*[−/−]; Jackson #004742), *Casp1*[−/−]*Casp11*[129mt/129mt] (referred to as *Casp1/11*[−/−])[64], *Gsdmd*[−/−][65], *Rag1*[−/−] (Jackson #0022216), *Prf1*[−/−] (Jackson #002407), *Casp7*[−/−] (Jackson #006237), *Nos2*[−/−] (Jackson #002609) and *Acod1*[−/−] (Jackson #029340) mice were used as indicated. All strains were maintained on 12/12 light/dark cycles, at 72+/− 2 °F, and under the humidity set point of 45%.

### Histology
For paraffin-embedded tissues, whole livers were harvested from euthanized mice and placed in 50 mL conical tubes containing 10% buffered formalin (VWR Cat. No. 16004-128). Tubes were inverted and swirled every other day for a minimum of three days to allow for full penetration of formalin into the tissue. Once fixed the livers were placed in tissue cassettes and transferred to the Histology Research Core at the University of North Carolina at Chapel Hill for embedding, cutting, slide mounting, and staining. The Histology Research Core performed all H&E and Sirius Red staining and serial sectioning of paraffin-embedded tissues. For frozen fixed tissues, mice were whole-body perfused through the heart with 2% paraformaldehyde (diluted down in PBS from 16%; VWR Cat. No. 15710-S), then livers were harvested and placed in 2% paraformaldehyde for 24 hours at 4 °C, followed by 30% sucrose (Sigma-Aldrich Cat. No. S1888) in PBS for 48 hours at 4 °C. After incubating in 30% sucrose, livers were placed in OCT media (Sakura Cat. No. 4583) at room temperature for 4 hours, then inserted into frozen tissue cassettes filled with OCT media and frozen at −80 °C. Frozen tissues were sectioned using a cryostat (CryoStar NX70). Pathologic analysis was performed with oversight from a board-certified veterinary pathologist (S.A.M.).

### Tissue processing for bacterial plating
Whole livers were harvested at indicated timepoints and placed into 7 mL homogenizer tubes (Omni International Cat. No. 19-651) containing 3 mL sterile PBS and one, 5 mm stainless steel bead (QIAGEN Cat. No. 69989). Spleens and single granulomas were harvested at indicated timepoints and placed into an 1 mL homogenizer tubes (Fisher Brand Cat. No. 14-666-315) containing 1 mL sterile PBS and one, 5 mm stainless steel bead. All tissues were homogenized on a Fisherbrand Bead Mill 24 Homogenizer for livers, and a Retsch MM400 Homogenizer for spleens and single granulomas. After homogenization, tissue lysates were serially diluted 1:5 in sterile PBS and plated on

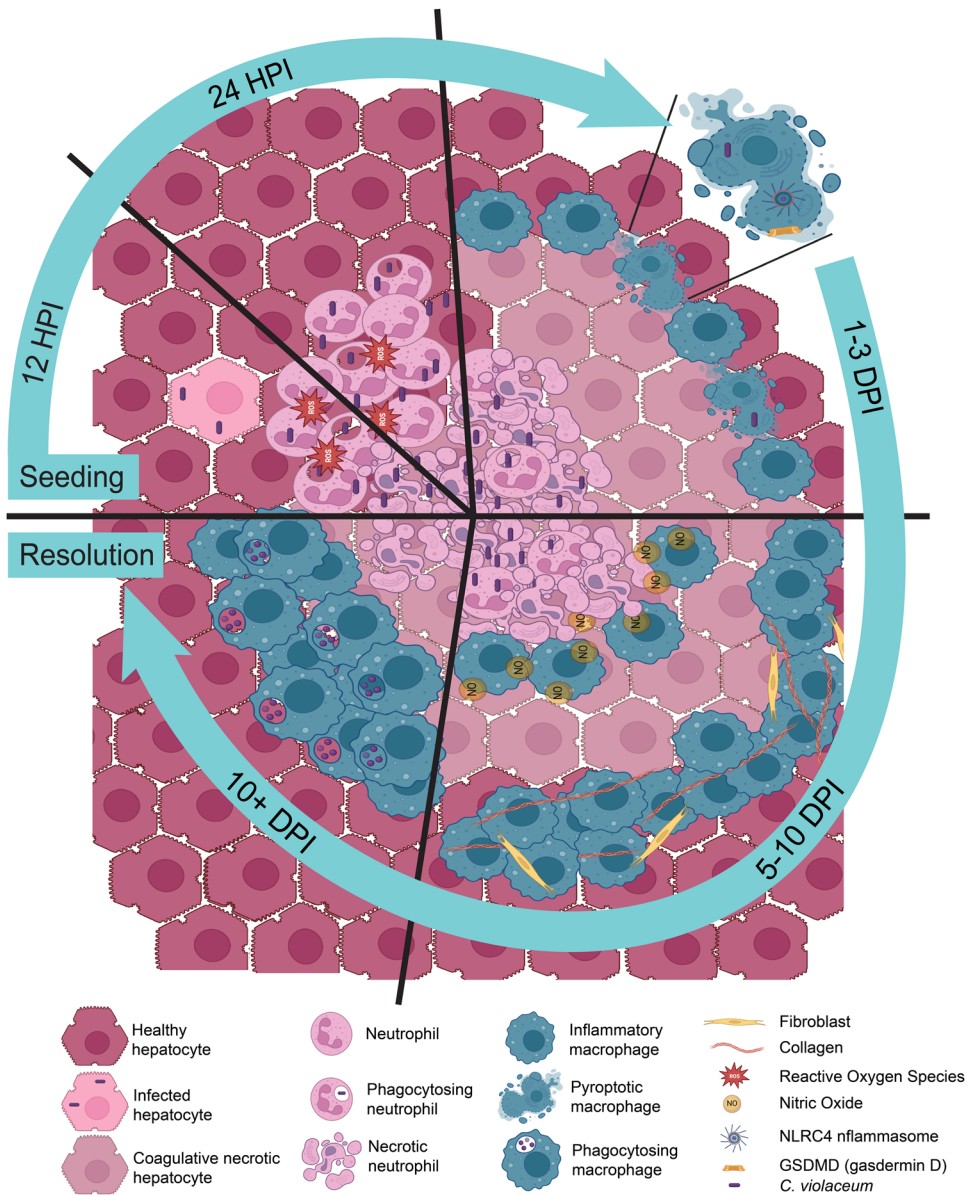

**Fig. 8 | Evolution and resolution of the *C. violaceum*-induced granuloma.** During the first hours of *C. violaceum* infection in the liver of WT mice, a single bacterium infects a hepatocyte. Infected hepatocytes are visible 12 hpi, in which bacteria rapidly replicate with a 1.5 hour doubling time. Shortly thereafter at 24 hpi, a neutrophil swarm appears and forms a microabscess. The ROS produced by neutrophils helps to slow the replication rate to a 14.5 hour doubling time but fails to clear the infection. At 3 dpi, macrophages arrive and begin to organize at the periphery of the microabscess, and bacterial replication halts. By 5 dpi this macrophage zone is pronounced, defining the pathology as a granuloma. Macrophages continue to form a barrier around the necrotic core that is composed mostly of *C. violaceum* and debris from dead neutrophils. Pyroptosis is critical to maintain the integrity of the granuloma, likely by killing any macrophage that becomes infected by *C. violaceum*, thus closing the macrophage replicative niche. Using nitric oxide, macrophages kill *C. violaceum* at the border of the necrotic core of the granuloma as early as 7 dpi. Once the bacteria are killed, macrophages can then infiltrate the inner layers of the granuloma to efferocytose the necrotic debris and dead bacteria. At later times when all the *C. violaceum* are dead, bacterial antigens can now be seen within the outer macrophage zone. Finally, the granuloma pathology is fully resolved and the liver returns to homeostasis. All this is accomplished by the innate immune system without the need for adaptive immune help during the primary infection.

BHI agar. Plates were incubated overnight at 37 °C and colony-forming units counted.

## Immunohistochemistry and Immunofluorescent staining

For immunohistochemistry of paraffin-embedded tissue, slides with 5 μm thick tissue sections were washed in xylene (Epredia Cat. No. 9990501) for 15 minutes, 100% ethanol (Sigma-Aldrich Cat. No. E7023) three times for 3 minutes each, 95% ethanol two times for 3 minutes each, 80% ethanol for 3 minutes, then washed in ddiH2O for 8 minutes to rehydrate the tissues. For antigen retrieval, slides were incubated in sodium citrate buffer (2.94 g Tri-Sodium Citrate, dH20 1 L, mix to dissolve, pH to 6.0 with 1 N HCl, add 0.05 mL Tween-20) and placed in a pressure cooker (Instant Pot Ultra) on the high pressure setting for 12 minutes. Slides were cooled for a minimum of 20 minutes in a slide rack at room temperature and then washed in TBS-T 1× (100 mL 10× TBS, 900 mL ddiH20, 1 mL Tween-20) for 1 minute. A pap pen was used to make a hydrophobic border around the tissues and then tissues were blocked with normal goat serum blocking solution, 2.5% (Vector Laboratories S-1012-50) at room temperature for 30 minutes. Tissues were washed with TBS-T 1×, then tissues were blocked with an Avidin/

Biotin Blocking Kit (Vector Laboratories SP-2001) following manufacturer protocol, and then washed in TBS-T 1× again. Primary antibodies for IHC: *C. violaceum*, 1:2000 (rabbit antisera custom generated by Cocalico Biologicals); Ly6G, 1:300 (Biolegend Cat. No. 127601); CD68, 1:200 (Abcam Cat. No. ab125212); CD3, 1:500 (Abcam Cat. No. ab5690); GFP, 1:200 (Invitrogen Cat. No. A11122); antibodies were diluted in SignalStain antibody diluent (Cell Signaling 8112 L) and incubated overnight at 4 °C in a humidity chamber. Then slides were washed with TBS-T 1× for 1 minute three times, then blocked for 10 minutes with 3% hydrogen peroxide (Sigma Cat. No. H1009) to block endogenous peroxidase activity. Slides were washed three times for 1 minute, then incubated with a secondary antibody polymer (Cell Signaling-Signal Stain boost, HRP anti-rabbit, 8114; ImmPRESS® HRP Goat-Anti-Rat IgG Polymer Detection Kit, Peroxidase, Cat. No. MP-7404-50). Slides were washed three times for 5 minutes, then stained with a DAB Substrate Kit, Peroxidase (HRP), (3,3'-diaminobenzidine; Vector Labs SK-4100) for 30–45 seconds. Slides were then counter-stained with Harris Modified Hematoxylin (Epredia Cat. No. 72704) or Meyers Modified Hematoxylin (Abcam Cat. No. ab220365) for 10-30 seconds. Hematoxylin was blued using running tap water for 1 minute. Slides were dehydrated in 50% ethanol for 2 minutes, 70% ethanol for 2 minutes, 100% ethanol 2 times for 2 minutes each, then xylene 2 times for 5 minutes each. Slides were then covered with Permount mounting medium (Fisher Chemical SP15-100) and covered with a coverslip.

For Immunofluorescence of frozen tissue, slides with 5 μm thick tissue sections were brought to room temperature for 15–20 minutes and a pap pen was used to draw a hydrophobic barrier around the tissue. Tissues were blocked with 2% Fc block (BD Pharmingen Cat. No. 553142) with 5% normal goat serum in 1× PBS at room temperature for 1 hour. Slides were washed for 1 minute in 1× PBS and then incubated with primary antibody overnight at 4 °C in a humidity chamber. Primary antibodies for IF: Ly6G Alexa Fluor 647, 1:100 (Biolegend Cat. No. 127610), CD68 Alexa Fluor 488, 1:100 (Abcam Cat. No. ab201844), iNOS Alexa Fluor 568, 1:100 (Abcam Cat. No. ab209595), e-cadherin, 1:200 (Abcam Cat. No. ab15148). Slides were then washed in 1× PBS three times for 5 minutes each and incubated with a secondary antibody (Invitrogen Goat-anti-rabbit 594, 1:1000, Cat. No. A32740) at room temperature for 1 hour. Slides were washed in 1× PBS three times for 5 minutes each, mounted using Fluoroshield™ with DAPI (Sigma-Aldrich F6057), and covered with a coverslip.

## Histology image capture
All histology images were captured using a Keyence all-in-one Fluorescent Microscope BZ-X800/BZX810. For histology image stitching and IF analysis, the BZ-X800 analyzer was used.

## Neutrophil depletion
24 hours prior to infection, mice were IP injected with 0.2 mg of either anti-mouse Ly6G/Ly6C antibody (BioXCell Cat. No. BE0075) or the isotype control anti-keyhole limpet hemocyanin antibody (BioXCell Cat. No. BE0090). At the zero timepoint, mice were given a repeat dose of the appropriate antibody and were infected with $1 \times 10^4$ CFU *C. violaceum*, as previously described. Livers were taken for histology at 24 hours post infection, as previously described.

## 10x genomics visium
Infected mouse liver tissues were harvested at various time points, embedded in OCT (Sakura Cat. No. 4583), and frozen. Frozen livers were cut to optimal section thickness and placed on a Tissue Optimization Slide to determine permeabilization conditions. Tissues were then placed within a 6.5mm² field on an expression slide that contained 5000 barcoded probes. The tissues were then fixed and stained with Hoechst and Eosin then permeabilized to release mRNA which binds to spatially barcoded capture probes, allowing for the capture of gene expression information. Captured mRNA from the slide surface was denatured, cleaved, and transferred into a PCR tube. From there, the cDNA was amplified, and standard NGS libraries were prepared. Adapters were ligated to each fragment followed by a sample index PCR. The libraries were sequenced to an average of 50,000 reads/probe on a paired-end, dual-indexed flow cell in the format of 28x10x10×90. Data was then uploaded to analysis packages for visualization.

Visium spatial data was analyzed using 10xGenomincs Space Ranger software and visualization through Loupe Browser[66,67]. Secondary statistical analysis was performed using a Seurat package in R[68]. Linear dimensional reduction was performed to calculate principal components using the most variably expressed genes in the spatial data. Spots were grouped into an optimal number of clusters for de novo cell type discovery using Seurat's FindNeighbors() and FindClusters() functions, and graph-based clustering approaches with visualization of spots were achieved through the use of manifold learning technique UMAP (Uniform Manifold Approximation and Projection)[69]. SpatialDimPlot() function in Seurat overlays the clustering results on the image to give combined plots of the expression data and the histology images. Additional downstream analyses include examining the distribution of a priori genes of interest, closer examination of genes associated with spot clusters, and the refined clustering of spots in order to identify further resolution of cell types. Spatial sequencing data is provided in Supplementary Data 1 and relevant code is provided in Supplementary Data 2.

## Module-based analysis of spatial sequencing data
To understand the cellular composition of each cluster, we leveraged two published single-cell RNA-seq datasets[31,32] from mouse liver and utilized the markers for immune cells and non-hematopoietic cells to calculate the module scores of different major cell types (Table S4E). The module score was generated as previously described[70] using the AddModuleScore function from the Seurat R package version 4[71]. Results were visualized using the ggplot2 R package and used to track the approximate relative abundance of unique cell types or specific genes at each infection time point. Abbrev. Key for reference: NC = Necrotic Core, CN = Coagulative Necrosis, M = Macrophage, EC = Endothelial Cell, OG = Outside Granuloma, Hep = Hepatocyte. Codes used are provided as Supplementary Data 2.

## Flow cytometry
Whole livers were harvested at the indicated timepoints post infection. Briefly, mice were euthanized according to IACUC guidelines, followed by whole-body perfusion with PBS (Gibco™, Cat. No. 14190-144). Whole livers were harvested and minced on ice using scissors, followed by incubation in digestion buffer (100 U/mL Collagenase Type IV (Gibco™, Cat. No. 17104019) prepared in DMEM (Gibco™, Cat. No. 11885-084), supplemented with 1 mM CaCl₂ and 1 mM MgCl₂,) at 37 °C for 40 minutes with intermittent vortexing. Digested tissues were mechanically homogenized through a Falcon® 40 μm cell strainer (Corning, Cat. No. 352340) to remove the majority of hepatocytes, and washed twice with RPMI (Gibco™, Cat. No. 11875-093) supplemented with 1% FBS (CPS Serum, Cat. No. FBS-500-HI) and 1× penicillin/streptomycin (Gibco™, Cat. No. 15140-122), followed by centrifugation in an Eppendorf® centrifuge (model 5810 R) at 1200 rpm (290 g) for 8 minutes at room temperature. Leukocytes were further isolated using a Percoll® gradient: samples were resuspended in 45% Percoll® (GE Healthcare, Cat. No. 17-0891-01), prepared in DMEM + 1.5 M NaCl), with an 80% Percoll® (prepared in PBS + 1.5 M NaCl) underlay, and spun for 20 minutes at 2000 rpm (805 g), room temperature, with no brake. Following collection of the leukocyte layer at the gradient interface, samples were washed twice, as before, and red blood cells were lysed with 1× RBC Lysis Buffer (eBioscience, Cat. No. 00-4333-57, according to product manual). Cells were washed and counted using trypan blue. $1 \times 10^6$ cells from each liver were stained for various cell markers: Live-or-Dye™ fixable viability dye in APC-Cy™7 (Biotium, Cat. No. 32008,

according to the product manual), Mouse BD Fc Block™ (BD Biosciences, Cat. No. 553142, according to product manual) rat anti-mouse Ly6G in BV421™ at 1:300 for 30 minutes (BD Horizon™, Cat. No. 562737), and finally, rat anti-mouse CD68 in FITC at 1:300 for 30 minutes (BioLegend, Cat. No. 137005) using Intracellular Fixation & Permeabilization Buffer (eBioscience, Cat. No. 88-8824-00, according to the product manual). Cells were acquired on a BD LSRFortessa X-20 Cell Analyzer (Duke Flow Cytometry Core Facility), and analyzed using FlowJo (for Windows, version 10.7.1).

#### Quantification of relative *Nos2* expression

Granulomas were excised from infected livers and placed into 1 mL homogenizer tubes (Fisher Brand Cat. No. 14-666-315) containing 1 mL sterile PBS and one, 5 mm stainless steel bead (QIAGEN Cat. No. 69989). Nucleic acids were isolated from excised granulomas by homogenizing in TRIzol (Invitrogen, Cat. No. 15596026) for 10 minutes. After homogenization, RNA was isolated by adding chloroform to a final ratio of 1:5 chloroform:TRIzol, centrifuging, mixing the aqueous layer with 70% EtOH, and then applying to Qiagen RNeasy columns (according to the product manual) to isolate RNAs longer than 200 bp. RNA quality was evaluated by 1% bleach gel[72]. cDNA was generated from 1 ug of RNA using Oligo dT(12-18) primers (Invitrogen, 18418012) and SuperScript II reverse transcriptase (Invitrogen, 18064022). Resultant cDNA at 1:100 dilution was subject to qPCR performed using iQ SYBR Green Supermix (BioRad, Cat. No. 1708880) on a QuantStudio 3 Real-Time PCR System (Applied Biosystems, A28567). Primers for *Nos2* (NM_010927.4) and *B2m* (NM_009735.3) were designed using NCBI PrimerBlast[73]. *Nos2* primers were designed to span exons 12 and 13 of *Nos2*, where the calmodulin-binding domain was replaced with a neomycin cassette to generate the *Nos2* knockout mice[74]. *Nos2* was amplified using primers NOS2_F (TACCAGATCGAGCCCTGGAAGA) and NOS2_R (AGCAAAGAACACCACTTTCACCA), and *B2m* was amplified using primers B2M_F (TGTATGCTATCCAGAAAACCCCT) and B2M_R (AGCAAAGAACACCACTTTCACCA). *Nos2*$^{-/-}$ mice did not amplify any product with primers NOS2_F and NOS2_R by agarose gel and melt curve analysis. Quantification of relative expression of NOS2 to B2M was determined using the ΔΔCt method[75]. At least three biological replicates were sampled for each treatment, and three technical replicates were run for each condition.

#### Total nitrate and nitrite assay of *C. violaceum* granulomas

Blood from indicated mouse strains was harvested via heart puncture and centrifuged at $10,000 \times g$ for 6 minutes in serum separation tubes (BD Microtainer Blood Collection Tubes Cat. No. 365967). Serum was filtered through a Microcon-10kDa Centrifugal Filter Unit with Ultracel-10 membrane (Millipore Cat. No. MRCPRT010) per product manual. Filtered serum was diluted 1:8 in assay buffer and measured for total nitrate and nitrite using Nitrate/Nitrite Colorimetric Assay Kit (Cayman Chemical Cat. No. 780001). Absorbance was measured at 540 nm using a BioTek Synergy H1 Microplate Reader, and total nitrate and nitrite calculated from absorbance according to product manual.

#### Percent inflammation measurements

2× stitched images of whole liver were taken using a Keyence Microscope (BZ-X800). A control measurement was taken at this time using the internal Keyence measurement analysis tool to determine um^2 of control square. Images were then loaded into Microsoft Windows 10 Paint (version21H2) to trace both the whole liver and regions of inflammation. Traced images were loaded into Image J (win-64 version 1.53) and area measured using the tracing tool. Image J areas were converted into true um^2 area using the control measurement values. Percent inflammation was determined by dividing the total area of inflammation by the total liver area. For the frequency distribution histogram, the area of each individual lesion was measured and calculated against the total area of the liver section to determine the percent area inflammation per lesion.

Lesion percent areas were then compiled and arranged in histogram format to show the distribution of lesion size per mouse genotype. Each histogram bin has a width of 0.5% area inflammation.

#### Statistical analysis

All statistical analysis was performed with GraphPad Prism 9. Discrete data was first assessed for normal distribution using a Shapiro–Wilk normality test. Data with normal distribution was analyzed with either unpaired two-tailed *t* test (two groups) or a one-way ANOVA (3 or more groups). Discrete data that did not have a normal distribution, or ordinal data for histological scoring, was analyzed with a two-sided Mann–Whitney (two groups) or Kruskal–Wallis (three or more groups). Experiments with 2 factors were analyzed with a two-way ANOVA. Survival analyses were performed using a Kaplan–Meier survival curve with Bonferroni correction for multiple comparisons where appropriate. Briefly, the Bonferroni correction was performed by calculating the pairwise *P* values of each desired comparison according to the Kaplan–Meier analysis on Prism, and then calculating a corrected α value by dividing 0.05 by the number of comparisons being made. This corrected α value was compared to each pairwise *P* value, and if $P < \alpha$, the difference between the groups was determined to be statistically significant. Significance is denoted in the following order by Prism: n.s. $p > 0.05$, *$p < 0.05$, **$p < 0.01$, ***$p < 0.001$, ****$p < 0.0001$. Prism 9 is unable to provide more precise calculations beyond $p < 0.0001$.

#### Reporting summary

Further information on research design is available in the Nature Portfolio Reporting Summary linked to this article.

### Data availability

All relevant data are included in the Article or its Supplementary Information and Source Data files. Mouse Cell Atlas 1.0 dataset is publicly available at https://doi.org/10.1016/j.cell.2018.02.001. Source data are provided with this paper.

### Code availability

Code used for spatial sequencing analysis is included in Supplementary Data 2.

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

## Acknowledgements

We thank R. Flavell and R. Vance for sharing mice directly, or through The Jackson Laboratory. This work was supported by the following NIH grants: AI133236, AI139304, AI136920 (E.A.M.) AI133236-04S1 (C.K.H.); NSF Graduate Research Fellowship Program DGE 2139754 (T.J.A.). We thank the Histology Research Core Facility in the Department of Cell Biology and Physiology at the University of North Carolina, Chapel Hill NC for all paraffin-embedded histological services. We thank the Mole-cular Genomics Core at Duke University for processing and analyzing the spatial transcriptomics (10X Genomics Visium). We thank F. Lakhani, M. Artunduaga, S. Pereles, A. Vaidyanathan, R. Eplett, M. Mann, S. Redecke, L. Scarpelli, A. Bryan, and F. Lin for mouse colony upkeep. Figure S7 was created with BioRender.com under agreement number MZ25GM6ZU3.

## Author contributions

C.K.H. performed most of the experiments with T.J.A., M.A.D., F.W.S., C.A.L., V.I.M., and Z.P.B. performing some experiments. M.E.A per-formed flow cytometry and analysis. E.A.M. supervised the overall pro-ject. S.A.M. supervised histology analysis. H.N.L. and B.D.M. managed the mouse colony. C.Y. and C.J.B. performed the module-based analysis of spatial sequencing data; supervised by D.R.S. C.K.H. and E.A.M. wrote the paper.

## Competing interests

C.A.L. is employed by AbbVie. This article is composed of the authors' work and ideas and does not reflect the ideas of AbbVie. The remaining authors declare no competing interests.

## Additional information

**Peer review information** *Nature Communications* thanks Igor Kramnik, Robert Watson, and the other, anonymous, reviewer(s) for their con-tribution to the peer review of this work. A peer review file is available.

