## [Peer Review File · Nature Communications]

An innate granuloma eradicates an environmental pathogen using *Gsdmd* and *Nos2*Editorial Note: Parts of this Peer Review File have been redacted as indicated to maintain the confidentiality of unpublished data.

REVIEWER COMMENTS

Reviewer #1 (Remarks to the Author):

This manuscript presents a detailed exploration of local inflammatory reaction in mouse livers after infection with environmental bacteria *C.violaceum* (Cv). This is an opportunistic intracellular pathogen that can infect non-phagocytic cells. The authors discovered that after intraperitoneal infection in a mouse model, Cv develops lesions in the livers of the immunocompetent B6 mice that, eventually, control the infection. The manuscript painstakingly describes the trajectories of the hepatic lesions using standard histopathology, immunochemistry, and spatial transcriptomics. The authors demonstrate that mice with knockouts in key pyroptosis proteins Casp1/11 and Gsdmd, and Nos2 genes were highly susceptible to Cv infection and failed to develop organized lesions that resemble granulomas.

The authors elegantly demonstrate that individual lesions develop from single bacteria and develop through distinct phases. All the data is of high quality and histopathological observations are supported by quantification. The conclusions are supported with convincing experimental data. The overall description of the model suggests that it will be useful for further analysis of mechanisms of innate immunity and cell interactions within local granuloma-like structures. However, the manuscript needs substantial editing for brevity and clarity.

In its current form, it is too long and overloaded with details that blur the conceptual punch line. Also, comparisons of Cv lesions with more traditional *S.mansoni* and *M.tuberculosis* model need to be limited to one paragraph in the Discussion. It is appropriate to draw some conclusions, but it seems overambitious to develop a unifying concept of granulomas based on the presented model. Presented data rather support pathogen-specific differences. Instead, it would be more important to describe the stages of the Cv lesion progression from the perspective of predominant localization of the pathogen and the main effector cell types and mechanisms of resistance that unfold in a stage-specific manner.

It seems that lesion progression can be separated into necrosuppurative and granulomatous phases. It should be clarified where the bacteria reside at each stage and what host cells are permissive for the bacterial replication. It seems that during the first 3 days the bacteria are localized in neutrophils where it rapidly replicates. Based on the presented numbers it would be possible to calculate the Cv generation time at different stages and, perhaps, compare it to the replication rate in vitro. Also, it seems that pyroptosis pathway is critically important during the first 3 days, i.e. the neutrophil stage. It would be important to discuss which cells may undergo pyroptosis in this model and how this may affect the bacterial replication. A cartoon describing the stages and corresponding effector mechanisms would help the discussion.

Thus, the overall recommendation is to shorten the manuscript and reduce the number of descriptive panels in main figures; move some pathology and corresponding pathology report language to supplement. More concise and less descriptive text will be more accessible and interpretable. For example, Fig.6B compares bacterial loads in spleens and livers. Presenting it as a time course would better show trends for each organ and each genetic background.

Specific comments:

Introduction.

Please, shorten the Introduction by moving the text about TB granulomas to the discussion. Suffice it to say here that granulomas are universal, but diverse reactions to persistent stimuli, etc. Also, it would be important to expand the granuloma definition by mentioning that macrophages in granulomas undergo local differentiation and can be represented by specific phenotypes, such as epithelioid macrophages and foamy macrophages, and granulomas contain organized fibrotic tissue.

Results.

Using CFU data, would it be possible to calculate the Cv generation time in vivo at various stages and compare it to its replication rate in vitro?

Fig.2. Granulomas contain primarily mononuclear cells including epithelioid macrophages and often fibrotic tissue. Thus, the early lesions do not represent incipient granulomas. Neutrophil infiltrate and necrosis are not a universal step in granulomas formation. In this model granuloma formation becomes evident 5-7 dpi.

Fig3b – the bacteria and macrophages overlap at the periphery. Granuloma wall does not localize the bacteria and does not separate it from normal tissue. Are bacteria intracellular in macrophages at this stage?

Fig.4 - Is it possible that pyroptosis occurs primarily in neutrophils? At 3 dpi when these KO mice die there are very few macrophages in the lesions. Most of bacteria extracellular within the neutrophil-rich necrotic areas. Does it replicate at this stage extracellularly, or the neutrophil pyroptosis halts the bacterial replication?

Please correct and clarify:

“Some studies show that murine *M. tuberculosis* infection is exacerbated in the absence pyroptotic pathway mutants”

“... amorphous material locking defined cell borders and features, consistent with necrotic debris (Figure 1C)”.

Ischemia was observed in WT mice at 3 and 5 dpi (Figure 2A, arrowhead, and S3A) – no arrowhead indicating ischemia is visible in Fig.2A

Suppl.Fig.4G not shown.

Fig.5E – please include abbreviations in the Figure legend

Fig.7D shows increased inflammation in *Gsdmd* KO mice at 5 dpi, but most of these KO mice died before day 5 (Fig.4A). This needs to be reconciled in the text.

Reviewer #2 (Remarks to the Author):

This is an extension of the authors' previous work (Maltez et al. *Immunity* 2015), both have similar conclusions on the role of Casp1/11, Ncf1 and RAG1 in the host response to *C. violaceum* (Cv). This study shows that Gasdermin D and NOS2 are independently required for the formation of the necrotic lesions in Cv-infected mice. While the study was executed largely with high quality, some additional experiments are needed to link the current and the previous work. In addition, the definition of the Cv-infected lesions as granulomas is questionable.

Specific comments:

1.The definition of the necrotizing lesions as granuloma is debatable. As defined by the authors, “Granulomas often form around pathogens that cause chronic infections” and the granuloma response is “accomplished independently of adaptive immunity that is typically required to organize granulomas.” However, this is not case for Cv infection, where the bacterium is eliminated rapidly in immunocompetent hosts and formation of the lesion is independent of T and B cells. The lesions represent a predominantly neutrophil-driven pathology, this is different to the typical granulomas formed in response to persistent pathogens, which is enriched with macrophages. In the latter case, increase in neutrophils are frequently associated with failed immunity. Strongly advise to avoid the use of the term granuloma, these lesions are better called something else, eg, necrotic lesions.

2.The manuscript states that “We speculate that this failure of neutrophils to kill the bacteria is the primary problem that triggers.”. Considering neutrophils are the predominant player in the lesion formation, neutrophil depletion experiments should be performed to establish the role of the leukocytes in the resistance and lesion formation at the time of infection.

3.NOS2 up-regulation is known to mainly depend on IFNs. It is somehow surprising that NOS2 KO mice are highly susceptible to Cv infection, as the authors have shown previously *Ifng*^{-/-} mice are not more sensitive to the bacterial infection than WT mice. It would be critical to test if NK depletion affects iNOS expression and granuloma formation in infected WT and *RAG1*^{-/-} mice.

Similarly, images of granulomas in infected *Ifng*^{-/-} mice should be shown along with those of *NOS2*^{-/-} mice. Finally, the expression and type I IFNs and related ISGs should also be investigated.

4. Bacterial inoculant dose is not stated in the manuscript.

5. Pathological presentations, some of the H&E-stained images are not consistent. For example, images in Fig.1 are not consistent with others (c, i vs. l). Moreover, the image in Fig 1J in the Immunity paper shows minimal pathology in the infected WT mice. How reproducible is this 3-region lesion morphology?

Reviewer #3 (Remarks to the Author):

In the manuscript "An innate granuloma eradicates an environmental pathogen using *Gsdmd* and *Nos2*" by Harvest et al., the authors investigate the *in vivo* response to the environmental/opportunistic pathogen *Chromobacterium violaceum*. Using a mouse model of infection, they demonstrate that after inoculation, this bacterium stimulates granuloma formation that successfully "walls off" and clears the infection. They show that each granuloma arises from one bacterium that replicates in the middle of a neutrophil swarm. Restriction of bacterial replication coincides with the arrival of macrophages around the periphery and formation of a granuloma. Using *Rag*-deficient mice they find that adaptive immune cells are not required to form a restrictive granuloma. They go on to show that both *Nos2* and *Gsdmd* are required to maintain the integrity and restrictive nature of the granuloma. Overall, this study supports that *C. violaceum*-induced granuloma formation is orchestrated by the innate immune cells and that this structure is sufficient to eradicate infection.

The model developed by the authors can certainly be leveraged to understand how a proper granuloma functions to clear infections and may provide new insights into granuloma biology during infection with important pathogens such as *Mycobacterium tuberculosis*. The authors employ a nice range of *in vivo* techniques/approaches to explore numerous facets of granuloma biology. That said, there are a couple experiments that that would help bolster this infection as a bona fide granuloma model and the authors conclusions drawn in this manuscript.

The manuscript focuses on mouse genetics and granuloma morphology readouts to show convincingly that gasdermin D and iNOS play a role in forming/maintaining the granuloma to clear bacteria. However, as it stands, it is difficult to draw major conclusions beyond that these factors are required *in vivo*. Both GSDMD and iNOS have shown to play both a role in limiting IL-1b production during *in vivo* infection. iNOS and GSDMD have ALSO shown to directly restrict intracellular bacteria replication (i.e. directly kill intracellular bacteria). Thus, additional experimentation would help to better understand/refine how both iNOS and GSDMD participate in the context of *Chromobacterium violaceum* infection and the granuloma.

For example:

(1) Does loss of *Nos2*/*Gsdmd* impact IL-1b production *in vivo* (in the granuloma)? Are there correlates of protection? e.g. Measure IL-1b (and other important cytokines) within the granuloma in wild-type versus *Nos2*/*Gsdmd* deficient mice at a time point where there are not significantly different bacterial loads.

(2) Is *Nos2*/*Gsdmd* responsible for restricting intracellular bacterial replication in macrophages *ex vivo*?

(3) Does iNOS and GSDMS play a role in cell death and/or release of IL-1b during infection? e.g. Measure IL-1b and cell death during infection in *Nos2*/*Gsdmd*/caspase-1 deficient BMDMs *ex vivo*. Again, it would be nice to determine the role of these iNOS and GSMDM in either direct killing or cell/death and IL-1b release.

(GSDMD also plays a very different role in the two major cell types present in the granuloma (macrophages and neutrophils) so experiments such as these might also begin to tease out the role of *Nos2*/*Gsdmd* in macrophages versus neutrophils.)

The data really suggests that *Chromobacterium violaceum* replicative niche might be a neutrophil. It might be challenging but measuring replication in neutrophils ex vivo might help better define the replicative niche in vivo.

I'm not sure the state of genetics for *Chromobacterium violaceum* but is it possible to make a TTSS mutant and determine whether it is required for granuloma formation?

Minor:

There are a couple of missing italics throughout the manuscript

The pics are beautiful but the ratio of the pic to font size (mostly in the graphs) are a little off such that it makes it a little challenging for to read some of the graphs.

REVIEWER REBUTTAL

We thank the reviewers for their time and effort in reviewing our manuscript, and for the overall positive reception.

Reviewer #1 (Remarks to the Author):

“This manuscript presents a detailed exploration of local inflammatory reaction in mouse livers after infection with environmental bacteria *C.violaceum* (Cv). This is an opportunistic intracellular pathogen that can infect non-phagocytic cells. The authors discovered that after intraperitoneal infection in a mouse model, Cv develops lesions in the livers of the immunocompetent B6 mice that, eventually, control the infection. The manuscript painstakingly describes the trajectories of the hepatic lesions using standard histopathology, immunochemistry, and spatial transcriptomics. The authors demonstrate that mice with knockouts in key pyroptosis proteins *Casp1/11* and *Gsdmd*, and *Nos2* genes were highly susceptible to Cv infection and failed to develop organized lesions that resemble granulomas.

The authors elegantly demonstrate that individual lesions develop from single bacteria and develop through distinct phases. All the data is of high quality and histopathological observations are supported by quantification. The conclusions are supported with convincing experimental data. The overall description of the model suggests that it will be useful for further analysis of mechanisms of innate immunity and cell interactions within local granuloma-like structures...”

We thank the reviewer for their time and effort in reviewing our manuscript, and for the overall positive reception.

“...However, the manuscript needs substantial editing for brevity and clarity. In its current form, it is too long and overloaded with details that blur the conceptual punch line...”

We have made multiple edits throughout the text to cut down details and make wording more concise. We also have moved an entire section that describes the other pathology (ischemia and coagulation) that occurs outside the granuloma structure to the supplement. We think these edits help in the flow and readability for a general audience.

“...Also, comparisons of Cv lesions with more traditional *S.mansoni* and *M.tuberculosis* model need to be limited to one paragraph in the Discussion. It is appropriate to draw some conclusions, but it seems overambitious to develop a unifying concept of granulomas based on the presented model. Presented data rather support pathogen-specific differences. Instead, it would be more important to describe the stages of the Cv lesion progression from the perspective of predominant localization of the pathogen and the main effector cell types and mechanisms of resistance that unfold in a stage-specific manner.”

We agree with the reviewer; the *C. violaceum*-induced granuloma does not create a unifying concept of granulomas. We think that this is the strength of the *C. violaceum*-induced granuloma is that it is fully operational. In the future, we hope to compare different granuloma model pathogens to discover what aspects of a non-resolving granuloma are broken. For example, *C. violaceum* is successfully contained by gasdermin D-driven pyroptosis, but there is evidence that *M. tuberculosis* inhibits/evades pyroptosis. It could be that this is why the *M. tuberculosis*-induced granuloma fails to resolve the infection. We have revised the introduction

and discussion to ensure that we do not give the impression that *C. violaceum* unifies all granuloma responses – instead, we emphasize its distinctiveness.

“It seems that lesion progression can be separated into necrosuppurative and granulomatous phases. It should be clarified where the bacteria reside at each stage and what host cells are permissive for the bacterial replication...”

We agree with the reviewer and have clarified in the text where the bacteria reside during each stage and what cell types are permissive during each stage.

“...It seems that during the first 3 days the bacteria are localized in neutrophils where it rapidly replicates. Based on the presented numbers it would be possible to calculate the Cv generation time at different stages and, perhaps, compare it to the replication rate in vitro...”

We thank the reviewer for this suggestion, the calculation reveals that *C. violaceum* replicates remarkably quickly. In BHI broth, *C. violaceum* has a 1 hour doubling time. During the early phase where probably most replication occurs in hepatocytes (6-24 hpi), the doubling time is 1.5 hours, remarkably similar to growth in broth; this phase ends as the neutrophil swarm appears at 24 hpi. The next phase is primarily characterized by the presence of neutrophils (24-72 hpi), where the doubling time slows significantly to 14.5 hours; this phase ends when macrophages arrive at 72 hpi. This data is added to the supplement (Supplement Figure S1c).

“...Also, it seems that pyroptosis pathway is critically important during the first 3 days, i.e. the neutrophil stage. It would be important to discuss which cells may undergo pyroptosis in this model and how this may affect the bacterial replication...”

Please see response below in “Specific comments”.

“...A cartoon describing the stages and corresponding effector mechanisms would help the discussion.”

We have added this as Figure S7.

“Thus, the overall recommendation is to shorten the manuscript and reduce the number of descriptive panels in main figures; move some pathology and corresponding pathology report language to supplement. More concise and less descriptive text will be more accessible and interpretable...”

We have made text edits through to reduce jargon (many of these are not marked with blue text as they are deletions). Our goal is to communicate with a broad audience, for example, researchers who study pyroptosis may not have a deep background in histologic pathology. These researchers are our target audience, and to reach them we do keep some descriptive text to make the paper accessible. We have removed extraneous text that is not essential, for example, we moved the section (Ischemia and Coagulation) to the supplement, which streamlines the manuscript.

“...For example, Fig.6B compares bacterial loads in spleens and livers. Presenting it as a time course would better show trends for each organ and each genetic background.”

For Figure 6B, we combined day 3 and 5 into one graph as these mice were infected at the same time. However, the 7 dpi mice were infected in a separate experiment. The conclusions we draw in the paper are valid and do not require us to repeat all three timepoints in one experiment. All the timepoints are representative of multiple experiments.

“Specific comments:
Introduction.

Please, shorten the Introduction by moving the text about TB granulomas to the discussion. Suffice it to say here that granulomas are universal, but diverse reactions to persistent stimuli, etc. Also, it would be important to expand the granuloma definition by mentioning that macrophages in granulomas undergo local differentiation and can be represented by specific phenotypes, such as epithelioid macrophages and foamy macrophages, and granulomas contain organized fibrotic tissue.”

We have removed the *M. tuberculosis* granuloma section from the introduction, as the reviewer suggests. We do discuss *M. tuberculosis* and other pathogens, in the discussion. We agree with the reviewer that granulomas morphology can be diverse, and we revised the text throughout to state this more clearly. We also added the different differentiation states macrophages can undergo during granuloma responses to the introduction. In the results we now note that the *C. violaceum*-induced granuloma macrophages are not epithelioid, foamy, or multinucleated.

“Results.

Using CFU data, would it be possible to calculate the *Cv* generation time in vivo at various stages and compare it to its replication rate in vitro?”

See above comment.

“Fig.2. Granulomas contain primarily mononuclear cells including epithelioid macrophages and often fibrotic tissue. Thus, the early lesions do not represent incipient granulomas. Neutrophil infiltrate and necrosis are not a universal step in granulomas formation. In this model granuloma formation becomes evident 5-7 dpi.”

We agree with the reviewer’s descriptions and definitions. We clarified the sequential pathology steps from microabscess (1-3 dpi), to a transitioning microabscess to granuloma (3-5 dpi), to mature granuloma (5-7 dpi). The text has been edited throughout to make this clearer.

“Fig3b – the bacteria and macrophages overlap at the periphery. Granuloma wall does not localize the bacteria and does not separate it from normal tissue. Are bacteria intracellular in macrophages at this stage?”

Indeed, at 14 dpi, the bacterial antigen staining overlaps with macrophage staining. At this timepoint, the vast majority of WT mice have no bacterial burdens (Figure 3f and S2a-b). Thus, this bacterial antigen staining represents dead bacteria within a resolving granuloma. In rare granulomas at late timepoints, we do observe live bacteria (Figure S2e). Such rare granulomas occur in livers where all the other granulomas have sterilized the bacteria (Figure S2e). These rare slow to clear granulomas continue to contain bacterial antigen staining in the core (Figure S2c). We edited the text to make this clearer.

“Fig.4 - Is it possible that pyroptosis occurs primarily in neutrophils? At 3 dpi when these KO mice die there are very few macrophages in the lesions. Most of bacteria extracellular within the neutrophil-rich necrotic areas. Does it replicate at this stage extracellularly, or the neutrophil pyroptosis halts the bacterial replication?”

This is an interesting point raised by the reviewer, and one that we are actively exploring. We have started to investigate which cell types require caspase-1 during *C. violaceum* infection using *Casp1^{fl/fl}* mice.

[REDACTED]

Therefore, dissecting the protective role of caspase-1 in different cell types will require significant additional research. We will publish this as a subsequent paper.

[REDACTED]

[REDACTED]

Please correct and clarify:

“Some studies show that murine *M. tuberculosis* infection is exacerbated in the absence pyroptotic pathway mutants” We agree. The prior discussion was too lengthy and convoluted in the discussion of *M. tuberculosis* and pyroptosis. We deleted a paragraph in the introduction and another in the discussion and the new discussion is more clear.

“... amorphous material locking defined cell borders and features, consistent with necrotic debris (Figure 1C).” This was a typographical error, we now changed ‘locking’ to ‘lacking’

“Ischemia was observed in WT mice at 3 and 5 dpi (Figure 2A, arrowhead, and S3A) – no arrowhead indicating ischemia is visible in Fig.2A” We added arrowheads.

“Suppl.Fig.4G not shown.” This was mislabeled, the correct callout is for Figure S4e

“Fig.5E – please include abbreviations in the Figure legend.” Added to figure legend.

“Fig.7D shows increased inflammation in *Gsdmd* KO mice at 5 dpi, but most of these KO mice died before day 5 (Fig.4A). This needs to be reconciled in the text.” The data shown are from *Gsdmd*^{-/-} mice that survived until this timepoint, which is clarified now in the text.

Reviewer #2 (Remarks to the Author):

“This is an extension of the authors’ previous work (Maltez et al. Immunity 2015), both have similar conclusions on the role of Casp1/11, Ncf1 and RAG1 in the host response to *C. violaceum* (Cv). This study shows that Gasdermin D and NOS2 are independently required for the formation of the necrotic lesions in Cv-infected mice. While the study was executed largely with high quality, some additional experiments are needed to link the current and the previous work...”

We thank the reviewer for their time and effort in reviewing our manuscript, and for the overall positive reception.

“...In addition, the definition of the Cv-infected lesions as granulomas is questionable.”

Specific comments:

1.The definition of the necrotizing lesions as granuloma is debatable...”

We can conclude with the upmost confidence the *C. violaceum*-induced pathology is a textbook example of a granuloma. One of our authors, Dr. Stephanie Montgomery, is a board-certified veterinary pathologist who has been deeply involved in this project since its inception for the last five and a half years. We also presented these slides to the veterinary pathologists at Duke University and North Carolina State University and they agreed with Dr. Montgomery’s assessment. The singular defining characteristic of a granuloma is the presence of organized macrophages. We show extensive data characterizing this defining macrophage zone and how it forms over the time during *C. violaceum* infection. The macrophage zone is mature at 5 dpi and this is the time when we define the pathology as a granuloma.

“...As defined by the authors, “Granulomas often form around pathogens that cause chronic infections” and the granuloma response is “accomplished independently of adaptive immunity that is typically required to organize granulomas.” However, this is not case for Cv infection, where the bacterium is eliminated rapidly in immunocompetent hosts and formation of the lesion is independent of T and B cells...”

We may have incorrectly given the reviewer the impression that granulomas *only* form around chronic infections. This is a common notion in the field because the pathogens that are most commonly studied fall into this category. In the introduction we take great efforts to make this clear by using qualifiers like “often” and “normally”, which we highlight in the introduction text in

blue font. This is important for this manuscript as we want to highlight the differences between the *C. violaceum*-induced granuloma and other more commonly studied pathogens. The pathological definition of a granuloma requires the presence of organized macrophages, but may or may not be characterized by chronic infection and adaptive immune cells.

“...The lesions represent a predominantly neutrophil-driven pathology, this is different to the typical granulomas formed in response to persistent pathogens, which is enriched with macrophages...”

We agree with the reviewer that when the lesion is predominantly neutrophil pathology (day 1-3), this is not a granuloma. We previously referred to this phase in vague terms, calling it a “lesion”. More accurate is to classify the day 1 lesion as a “microabscess”, and we revise the text to use this more accurate term. Perhaps our lack of precise terminology led to the impression that we were calling the 1-3 dpi pathology granulomas when that was not our intention.

When macrophages appear at day 3, the classification is difficult because the macrophage zone is just starting to form, and we call this an “early granuloma”. By day 5 through 14, the macrophage zone is very apparent and the defining cell type of the pathology, which defines the pathology at these timepoints as a “granuloma”. Because this granuloma begins with neutrophils, we can sub-categorize the granuloma as a pyogranuloma (or synonymously a suppurative granuloma).

“...In the latter case, increase in neutrophils are frequently associated with failed immunity. Strongly advise to avoid the use of the term granuloma, these lesions are better called something else, eg, necrotic lesions.”

We agree that other publications show that neutrophils are associated with failed immunity, for example, in the late phase of *M. tuberculosis* infection the presence of neutrophils correlates with a response that fails to clear the bacterium (e.g. doi 10.1038/s41385-020-0300-z). We think that a failure of neutrophils to kill bacteria will become a defining characteristic of granuloma-inducing bacterial pathogens. Interestingly, this failed neutrophil phase occurs only in the first 3 days of *C. violaceum* infection, whereas it occurs late in the adaptive immune phase of an *M. tuberculosis* infection. Later during *C. violaceum* infection, neutrophil numbers decrease, which correlates with the success of the granuloma in sterilizing the bacteria.

“2.The manuscript states that “We speculate that this failure of neutrophils to kill the bacteria is the primary problem that triggers.”. Considering neutrophils are the predominant player in the lesion formation, neutrophil depletion experiments should be performed to establish the role of the leukocytes in the resistance and lesion formation at the time of infection.”

We agree that this is an interesting avenue for exploration. We depleted neutrophils

Without neutrophils, the replication of *C. violaceum* within hepatocytes is more readily apparent (Reviewer Figure 2c), and the bacteria even appears to spread from hepatocyte to hepatocyte.

We also add new data to the paper showing that in the neutrophil-predominant phase from 1-3 dpi *C. violaceum* replication

slows from a 1.5 hour doubling time to a 14.5 hour doubling time (Figure S1c). The neutrophil depletion data will be published in a future manuscript focused on the nature of the neutrophil failure against *C. violaceum*.

3. NOS2 up-regulation is known to mainly depend on IFNs. It is somehow surprising that NOS2 KO mice are highly susceptible to Cv infection, as the authors have shown previously *lfn*^{-/-} mice are not more sensitive to the bacterial infection than WT mice. It would be critical to test if NK depletion affects iNOS expression and granuloma formation in infected WT and *RAG1*^{-/-} mice. Similarly, images of granulomas in infected *lfn*^{-/-} mice should be shown along with those of NOS2^{-/-} mice. Finally, the expression and type I IFNs and related ISGs should also be investigated.”

Indeed, this is the dogma in the field that NOS2 is primarily driven by IFN- γ . We previously showed that *lfn*^{-/-} mice have normal bacterial burdens at 3 dpi, which matches the lack of a phenotype for *Nos2*^{-/-} mice at 3 dpi. [REDACTED]

[REDACTED]

We will further investigate these phenotypes which will result in its own manuscript.

[REDACTED]

“4. Bacterial inoculant dose is not stated in the manuscript.”

We have indicated inoculant dose within the manuscript results text as well as each figure legend.

“5. Pathological presentations, some of the H&E-stained images are not consistent. For example, images in Fig. 1 are not consistent with others (c, i vs. l)..”

All H&E staining was performed by the UNC Research Histology Core. We understand that not all the H&E staining has identical intensity. Different experiments were stained at different times and if the hematoxylin is old for one experiment, and new for another, can result in different hues. Unfortunately, this is the nature of all histological staining. For any particular stain, slides within an experiment were processed on the same day.

“...Moreover, the image in Fig 1J in the Immunity paper shows minimal pathology in the infected WT mice. How reproducible is this 3-region lesion morphology?”

In the prior *Immunity* paper, we used a lower inoculum of 10^2 CFUs compared to 10^4 CFUs in the current paper. The low inoculum resulted in one or zero lesions per WT mouse liver, but many more in *Casp1/11^{-/-}* mice. At the time made us think that the lesions in WT mice were an

anomaly. In the pathology in the previous paper, the entire liver had zero lesions, so the picture shown in that paper was representative of that mouse, as there were no lesions to image. We had simply selected a random area of liver as representative. We now note this in the current manuscript “When a lower 10^2 CFU infectious dose is used, WT mice can show no lesions”. However, when we used a 10^4 CFU infectious dose, many lesions appear in WT mice, as shown in the current manuscript. At this dose, these are extremely reproducible, which we include in the manuscript by showing whole liver sections so the reader can see multiple granulomas in a single mouse and assess the reproducibility.

Reviewer #3 (Remarks to the Author):

In the manuscript “An innate granuloma eradicates an environmental pathogen using *Gsdmd* and *Nos2*” by Harvest et al., the authors investigate the *in vivo* response to the environmental/opportunistic pathogen *Chromobacterium violaceum*. Using a mouse model of infection, they demonstrate that after inoculation, this bacterium stimulates granuloma formation that successfully “walls off” and clears the infection. They show that each granuloma arises from one bacterium that replicates in the middle of a neutrophil swarm. Restriction of bacterial replication coincides with the arrival of macrophages around the periphery and formation of a granuloma. Using Rag-deficient mice they find that adaptive immune cells are not required to form a restrictive granuloma. They go on to show that both *Nos2* and *Gsdmd* are required to maintain the integrity and restrictive nature of the granuloma. Overall, this study supports that *C. violaceum*-induced granuloma formation is orchestrated by the innate immune cells and that this structure is sufficient to eradicate infection.

The model developed by the authors can certainly be leveraged to understand how a proper granuloma functions to clear infections and may provide new insights into granuloma biology during infection with important pathogens such as *Mycobacterium tuberculosis*. The authors employ a nice range of *in vivo* techniques/approaches to explore numerous facets of granuloma biology. That said, there are a couple experiments that would help bolster this infection as a bona fide granuloma model and the authors' conclusions drawn in this manuscript.”

We thank the reviewer for their time and effort in reviewing our manuscript, and for the overall positive reception.

“The manuscript focuses on mouse genetics and granuloma morphology readouts to show convincingly that gasdermin D and iNOS play a role in forming/maintaining the granuloma to clear bacteria. However, as it stands, it is difficult to draw major conclusions beyond that these factors are required *in vivo*. Both GSDMD and iNOS have shown to play both a role in limiting IL-1b production during *in vivo* infection. iNOS and GSDMD have ALSO shown to directly restrict intracellular bacteria replication (i.e. directly kill intracellular bacteria). Thus, additional experimentation would help to better understand/refine how both iNOS and GSDMD participate in the context of *Chromobacterium violaceum* infection and the granuloma.”

We have added several new pieces of data to provide additional mechanistic insights into the independent roles of gasdermin D and iNOS. “To quantitate the defect in the overall granuloma response, we measured the area of total active inflammation in the liver section as a percentage of the total tissue area. This included functional granulomas in WT mice, defective granulomas in the various knockout mice, as well as newly seeded microabscesses seen in the *Casp1/11*^{-/-} and *Gsdmd*^{-/-} mice. Both *Casp1/11*^{-/-} and *Gsdmd*^{-/-} mice have increased inflammation as well as increased numbers of inflammatory regions within the liver at 3 dpi (Figure S6h, S6i, S6j, and

S6k), which correlates with increased bacterial burdens (Figure 4b, 4c). *Gsdmd*^{-/-} mice have increased inflammation as well as a higher number of small inflammatory regions compared to WT at 5 dpi, akin to 1 dpi microabscesses (Figure 7d and 7e). This data suggests that pyroptosis deficient mice have a double failure during in containing *C. violaceum* infection. First, the 'budding' morphology indicates a failure of granuloma containment of *C. violaceum* into the local tissue. Second, the small microabscesses may be due to bacterial dissemination from the spleen, first seen at 3 dpi (Figure S3j and S3k).

At the same 3 dpi timepoint, *Nos2*^{-/-} mice did not have increased inflammation, inflammatory regions, or burdens compared to WT mice (Figure S6h, S6i, S6j, S6k and 6b). It was not until 5 dpi that the surviving *Nos2*^{-/-} mice have increased inflammation (to a similar degree as those *Gsdmd*^{-/-} mice that survived until this timepoint), as well as increased burdens (Figure 7d and 6b). We had selection bias for the knockout mice that survived until the 5dpi timepoint, we hypothesize the mice that died had worse pathology (Figure 7d). When we quantitated the inflammatory regions at 5 dpi, we saw *Nos2*^{-/-} mice had a similar number of inflammatory regions as WT (Figure S6l) but these regions are larger with the median region size greater than WT (Figure 7e). Additionally, at 5 dpi *Nos2*^{-/-} mice have a higher percent of all regions larger than three percent total area compared to WT (Figure 7e; brackets). This results in a greater total area of inflammation (Figure 7d). All this data suggests mice deficient in iNOS have a granuloma defect that allows local spread in the liver.”

“For example:

(1) Does loss of *Nos2/Gsdmd* impact IL-1b production in vivo (in the granuloma)? Are there correlates of protection? e.g. Measure IL-1b (and other important cytokines) within the granuloma in wild-type versus *Nos2/Gsdmd* deficient mice at a time point where there are not significantly different bacterial loads.”

We investigated the role of IL-1β in this granuloma defense response. We found that IL-1β knockout mice form normal granulomas with all the normal architectural layers that are seen in WT mice. Moreover, *Il1b*^{-/-} mice survive the infection, indicating that this cytokine is not important for the granuloma response (new data Figure S6a and S6b). Because there is no phenotype in vivo for IL-1β, we can conclude that iNOS and gasdermin D act independently of this cytokine.

“(2) Is *Nos2/Gsdmd* responsible for restricting intracellular bacterial replication in macrophages ex vivo?”

We have infected bone marrow derived macrophages with *C. violaceum* in vitro. Macrophages are remarkably efficient in detecting *C. violaceum* via NLRC4 detection that activates caspase-1. Even a low MOI of 5 results in 60% pyroptosis within 1 hour (new data Figure S6f). *Nos2*^{-/-} macrophages retain the ability to undergo pyroptosis. Therefore, *C. violaceum* cannot replicate intracellularly in macrophages in vitro because the macrophages are dead. We think the same is true in vivo. This again separates the functions of iNOS and gasdermin D.

“(3) Does iNOS and GSDMS play a role in cell death and/or release of IL-1b during infection? e.g. Measure IL-1b and cell death during infection in *Nos2/Gsdmd/caspase-1* deficient BMDMs ex vivo. Again, it would be nice to determine the role of these iNOS and GSMDM in either direct killing or cell/death and IL-1b release.”

We now examine whether iNOS is required for gasdermin D function and IL-1β release in vitro. *Nos2*^{-/-} macrophages remain competent to release IL-1β and undergo pyroptosis in response to *C. violaceum* infection (new Figure S6c and S6d). Thus, again, IL-1β does not seem to be the mechanism of action for iNOS or gasdermin D.

“(GSDMD also plays a very different role in the two major cell types present in the granuloma (macrophages and neutrophils) so experiments such as these might also begin to tease out the role of *Nos2/Gsdmd* in macrophages versus neutrophils.)”

This is an interesting point raised by the reviewer, and one that we are actively exploring. We have started to investigate which cell types require caspase-1 during *C. violaceum* infection using *Casp1^{fl/fl}* mice.

[REDACTED]

Therefore, dissecting the protective role of caspase-1 in different cell types will require significant additional research. We will publish this as a subsequent paper.

“The data really suggests that *Chromobacterium violaceum* replicative niche might be a neutrophil. It might be challenging but measuring replication in neutrophils *ex vivo* might help better define the replicative niche *in vivo*.”

We agree that the role of neutrophils and the possibility of replication inside neutrophils is an interesting avenue for exploration. We depleted neutrophils and assessed [REDACTED] d microabscess formation (Reviewer Figure 2).

[REDACTED] Without neutrophils, the replication of *C. violaceum* within hepatocytes is more readily apparent (Reviewer Figure 2), and the bacteria even appears to spread from hepatocyte to hepatocyte.

[REDACTED] We also add new data to the paper showing that in the neutrophil-predominant phase from 1-3 dpi *C. violaceum* replication slows from a 1.5 hour doubling time to a 14.5 hour doubling time (Figure S1c). The neutrophil depletion data will be published in a future manuscript focused on the nature of the neutrophil failure against *C. violaceum*.

“I’m not sure the state of genetics for *Chromobacterium violaceum* but is it possible to make a TTSS mutant and determine whether it is required for granuloma formation? ”

The importance of the CPI1 T3SS has been published by other labs previously (doi 10.1111/j.1365-2958.2010.07248.x), and we confirm their data that a T3SS mutant ($\Delta cilA$) cannot colonize the liver of WT mice (Reviewer Figure 4). There were no visible lesions on the livers of mice infected with the $\Delta cilA$ mutant.

[REDACTED]

“Minor:

There are a couple of missing italics throughout the manuscript”

We re-read the paper and corrected all formatting error that we could find.

“The pics are beautiful but the ratio of the pic to font size (mostly in the graphs) are a little off such that it makes it a little challenging for to read some of the graphs.”

We increased the font sizes to use the size specified by the journal guidelines (Arial 7 point).

REVIEWERS' COMMENTS

Reviewer #1 (Remarks to the Author):

In the revised manuscript, Harvest et al. significantly improved the clarity of presentation and the discussion. There are several typos that need to be corrected.

The main improvement is a new Supplemental Fig.7 that nicely shows stages of the granuloma progression in their model. This cartoon would be better placed in the main text. An new experiment using neutrophil depletion, presented in the rebuttal as Reviewer Fig.2, nicely supports the stage delineation. It should be part of this manuscript, because it documents specific role of neutrophil at the very early stages of the granuloma formation. This finding would strengthen the manuscript conceptually and would be on general interest, since the early stages of the granuloma development are less studies in other models.

Reviewer #2 (Remarks to the Author):

The authors have clarified most of my concerns. However, I think the data associated with the role of neutrophils in resistance to the infection (neutrophil deletion study, Reviewer Figure 2.) should be included to substantiate the main conclusion.

Reviewer #3 (Remarks to the Author):

The authors did a great job addressing my concerns/critiques! I have no further issues. Nice Work!

“Reviewer #1 (Remarks to the Author):

In the revised manuscript, Harvest et al. significantly improved the clarity of presentation and the discussion. There are several typos that need to be corrected.”

We again thank the reviewer for their continued time and effort in assessing our manuscript. We have proofread the manuscript again and corrected typographical errors.

“The main improvement is a new Supplemental Fig.7 that nicely shows stages of the granuloma progression in their model. This cartoon would be better placed in the main text.”

We agree that this model figure will be very helpful for the reader and is better in the main text. Due to the image intensity of our figures, there is not space in the current 7 Figures to include this model figure, and we added it as Figure 8.

“An new experiment using neutrophil depletion, presented in the rebuttal as Reviewer Fig.2, nicely supports the stage delineation. It should be part of this manuscript, because it documents specific role of neutrophil at the very early stages of the granuloma formation. This finding would strengthen the manuscript conceptually and would be on general interest, since the early stages of the granuloma development are less studies in other models.”

We had included this data as a reviewer figure

[REDACTED]

Regardless, the conclusions that we draw in this manuscript are that the neutrophil swarm fails to kill the bacteria, and this remains true. We also repeated the staining of bacteria in neutrophil depleted mice, and this experiment was repeatable. Neutrophil depletion allows us to visualize the hepatocyte phase of the infection in greater length. We now include these experiments in Figure 1h and S1d.

“Reviewer #2 (Remarks to the Author):

The authors have clarified most of my concerns.”

We again thank the reviewer for their continued time and effort in assessing our manuscript.

“However, I think the data associated with the role of neutrophils in resistance to the infection (neutrophil deletion study, Reviewer Figure 2.) should be included to substantiate the main conclusion.”

We had included this data as a reviewer figure

[REDACTED]

Regardless, the conclusions that we draw in this manuscript are that the neutrophil swarm fails to kill the bacteria, and this remains true. We also repeated the staining of bacteria in neutrophil depleted mice, and this experiment was repeatable. Neutrophil depletion allows us to visualize the hepatocyte phase of the infection in greater length. We now include these experiments in Figure 1h and S1d.

“Reviewer #3 (Remarks to the Author):

The authors did a great job addressing my concerns/critiques! I have no further issues. Nice Work!”

We again thank the reviewer for their continued time and effort in assessing our manuscript.